# LiveResearchBench: A Live Benchmark for User-Centric Deep Research in the Wild

**Jiayu Wang**[1*], **Yifei Ming**[3*], **Riya Dulepet**[2], **Qinglin Chen**[3], **Austin Xu**[3], **Zixuan Ke**[3],
**Frederic Sala**[1], **Aws Albarghouthi**[1], **Caiming Xiong**[3], **Shafiq Joty**[3]
[1]University of Wisconsin-Madison      [2]Stanford University      [3]Salesforce AI Research

## Abstract

Deep research—producing comprehensive, citation-grounded reports by searching and synthesizing information from hundreds of live web sources—marks an important frontier for agentic systems. To rigorously evaluate this ability, four principles are essential: tasks should be (1) *user-centric*, reflecting realistic information needs, (2) *dynamic*, requiring up-to-date information beyond parametric knowledge, (3) *unambiguous*, ensuring consistent interpretation across users, and (4) multi-faceted and search-intensive, requiring search over numerous web sources and in-depth analysis. Existing benchmarks fall short of these principles, often focusing on narrow domains or posing ambiguous questions that hinder fair comparison. Guided by these principles, we introduce **LiveResearchBench**, a benchmark of 100 expert-curated tasks spanning daily life, enterprise, and academia, each requiring extensive, dynamic, real-time web search and synthesis. Built with over 1,500 hours of human labor, LiveResearchBench provides a rigorous basis for systematic evaluation. To evaluate citation-grounded long-form reports, we introduce **DeepEval**, a comprehensive suite covering both content- and report-level quality, including coverage, presentation, citation accuracy and association, consistency and depth of analysis. DeepEval integrates four complementary evaluation protocols, each designed to ensure stable assessment and high agreement with human judgments. Using LiveResearchBench and DeepEval, we conduct a comprehensive evaluation of frontier deep research systems, including single-agent web search, single-agent deep research, and multi-agent systems. Our analysis reveals current strengths, recurring failure modes, and key system components needed to advance reliable, insightful deep research. Our code is available at: `https://github.com/SalesforceAIResearch/LiveResearchBench`.

## 1 Introduction

Deep research is the process of addressing complex, open-ended questions that require long-horizon, multi-step reasoning and planning. It involves exploring hundreds of live web sources and synthesizing them into comprehensive, citation-grounded, and structured outputs such as reports (OpenAI, 2025). This process closely mirrors how human researchers work—iteratively seeking, validating, and integrating evidence. Building systems capable of this marks an important frontier in AI: moving LLMs from simple chatbots to independent problem-solvers. However, progress is bottlenecked by a benchmarking and evaluation crisis: Which tasks are truly representative? How should ambiguities in query formulation be resolved? How can we reliably measure comprehensiveness (breadth and depth), consistency, factuality, and citation accuracy in long-form outputs while safeguarding against data contamination (Nguyen et al., 2025)?

Evaluating agentic tasks is challenging in general, and deep research is particularly difficult to benchmark due to its open-ended nature. To address this, we first propose four key principles for benchmarking deep research systems: tasks should be (a) user-centric, reflecting realistic information needs; (b) dynamic, requiring up-to-date information beyond parametric knowledge;

---

*Equal contribution. Correspondence to: milawang@cs.wisc.edu, {yifei.ming, sjoty}@salesforce.com

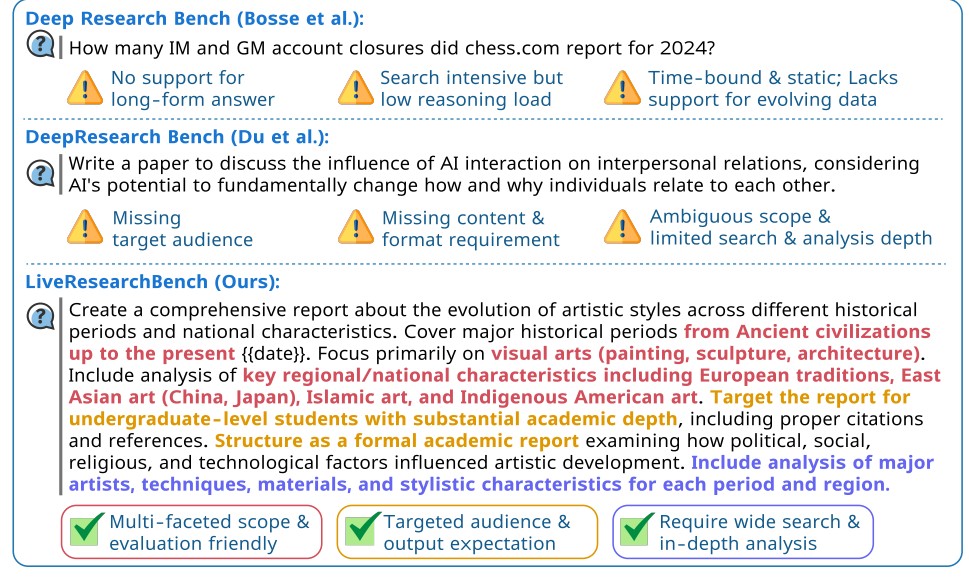

Figure 1: Comparison of LiveResearchBench with existing deep research benchmarks. `{{date}}` will be replaced by the evaluation date. LiveResearchBench is explicitly *user-centric*, *unambiguous*, and *time-varying*, featuring multi-faceted, search-intensive tasks across diverse domains.

(c) unambiguous, ensuring consistent interpretation across users; and (d) multi-faceted and search-intensive, requiring multi-hop search across diverse web sources and in-depth analysis. Recent benchmarks have introduced search-intensive tasks (Gou et al., 2025), but these are often static (and thus prone to contamination) (Bosse et al., 2025), domain-specific (Xu et al., 2025b; Patel et al., 2025), coarse-grained, or ambiguous (Du et al., 2025)—frequently omitting details such as the intended audience, required output format, or scope. Consequently, we still lack reliable tools to assess whether models are producing genuinely high-quality research or merely assembling plausible-sounding yet insufficient responses.

We address this gap with **LiveResearchBench**, a benchmark of 100 expert-curated queries paired with detailed checklists, constructed with over 1,500 hours of human labor. The tasks we benchmark span seven diverse domains (*e.g.,* science, business, healthcare) and ten categories such as market analysis, literature review, policy evaluation, and topic exploration. Each query is multi-faceted: it specifies scope, audience, and format whenever appropriate, and many require real-time search at evaluation time. For example, one task asks for a comprehensive report on the evolution of artistic styles "up to the present {{date}}," requiring breadth across historical periods, depth of analysis on techniques and materials, and presentation as a formal academic report for undergraduate students (Figure 1). Compared to prior work, tasks in LiveResearchBench are ***realistic and unambiguous***. It makes it possible to accurately gauge retrieval, synthesis, and reasoning in deep research.

While these queries capture more diverse scenarios, we must also ensure that the way we evaluate the resulting reports provides fine-grained insights into deep research agent capabilities. Evaluating the resulting long-form/open-ended reports poses unique challenges. For example, we cannot use simple string-matching approaches, while real-time queries imply that there is time-sensitive variation, without a fixed ground truths. Most critically, tasks are inherently multi-dimensional: we must assess coverage, reasoning, evidence use, and presentation quality. Human evaluation is costly and hard to scale, while naive LLM-as-a-judge setups produce inconsistent results.

To overcome these obstacles, we propose **DeepEval**, a comprehensive evaluation suite for research reports that spans both content-level and report-level metrics. Content-level metrics assess coverage and comprehensiveness, analysis depth, citation association, and factual accuracy. Report-level metrics evaluate presentation & organization, and logical & factual consistency. Each metric is paired with a tailored evaluation protocol (*e.g.*, checklist-based, pointwise, pairwise, or rubric-tree), selected to ensure stable model assessment and high agreement with human evaluations. This design enables scalable, fine-grained, and reliable evaluation of system performance in dynamic research settings, particularly for open-ended, long-form reports.

We summarize the key contributions as follows:

- We establish four core task design principles for deep research, grounded in user survey and comparative analysis, that ensure tasks are realistic, unambiguous, and evaluation-ready.

- We introduce **LiveResearchBench**, a benchmark of 100 expert-curated, checklist-validated tasks across diverse domains and categories, explicitly designed for dynamic, multi-faceted research.

- We propose **DeepEval**, a comprehensive evaluation suite for open-ended/long-form reports, covering six complementary dimensions and closely aligning with expert assessments.

- We conduct a comprehensive evaluation of 17 state-of-the-art open-sourced and proprietary single- and multi-agent systems. We reveal systematic vulnerabilities and error patterns, and that most models are yet incapable of writing insightful reports.

## 2 RELATED WORK

**Single- and Multi-Agent Deep Research Systems.** Equipping LLMs with tools—often referred to as agents—for grounded answers to complex queries has become a major research direction. Pioneering systems such as Deep Research using o3 (OpenAI, 2025) combined browsing and code execution and generate long reports. Many successors have since emerged (Xu & Peng, 2025; Java et al., 2025). Architectures generally fall into two broad categories. Single-agent systems place all tool-use decisions in one model. Examples include function-calling LLMs (Yang et al., 2025; Agarwal et al., 2025) and ReAct-style agents (Yao et al., 2022). Multi-agent systems (MAS) instead coordinate specialized roles such as planner, researcher, and writer in a predefined workflow. Proprietary multi-agent systems such as Grok 4 Heavy (xAI, 2025a) and Manus (Manus, 2025) remain undisclosed, while examples of open-source systems include Open Deep Research (Alzubi et al., 2025), DeerFlow (ByteDance, 2025), and OpenManus (Liang et al., 2025). Despite rapid progress, it is still unclear *which dimensions favor MAS vs. single-agent systems, what fundamental bottlenecks remain, and whether MAS truly deliver consistent gains* (Kapoor et al., 2025). The lack of standardized benchmarks and evaluation frameworks further prevents rigorous comparison. Our work directly addresses this gap by developing a comprehensive evaluation framework, enabling rigorous comparison of single- and multi-agent systems for dynamic and multi-faceted research tasks.

**Deep Research Benchmark.** Several benchmarks have been proposed for evaluating deep research. DeepScholarBench (Patel et al., 2025) targets related-work generation; DeepResearch Bench (Du et al., 2025) include 100 short open-ended questions; Deep Research Bench (Bosse et al., 2025), LiveDRBench (Java et al., 2025), and Mind2Web2 (Gou et al., 2025) primarily target closed-ended information-seeking tasks. While valuable, existing benchmarks exhibit some combination of the following limitations: they are domain-specific, restricted to short-form or closed-ended answers, and static, which hinders fair comparison across agents and prevents evaluation on evolving information. A comparison is shown in Figure 1. Recent efforts such as DeepResearchGym (Coelho et al., 2025) adopts short and static tasks, while ResearcherBench (Xu et al., 2025b) focuses narrowly on AI domain. In contrast, our LiveResearchBench provides *user-centric, unambiguous, and time-varying* tasks that are multi-faceted, search-intensive, and span diverse domains. A detailed discussion of benchmark comparison is provided in Appendix B.1.

**Evaluation of Long-form Answer.** Recent work has moved beyond simple reference-based metrics. Some benchmarks emphasize coverage and factual grounding (Laban et al., 2023; 2024; Huang et al., 2024; Xu et al., 2025a), such as PROXYQA (Tan et al., 2024) (sub-question coverage) and LongFact (Wei et al., 2024) (claim-level verification). Others target structure and overall quality, including LongEval (Wu et al., 2025) (plan-based vs. direct generation) and HelloBench (Que et al., 2024) (human-aligned protocols). However, these efforts typically isolate a single dimension. In contrast, DeepEval offers a *comprehensive suite*: report-level metrics (presentation, consistency) and fine-grained dimensions (depth, checklist-based coverage, citation traceability, rubric-tree accuracy). Crucially, DeepEval is *reliable and human-aligned*: instead of holistic scores, we use structured checklists (e.g., factual errors as atomic tests) and aggregate results, yielding stable grades that closely track expert judgments. A detailed discussion of evaluation methods for open-ended long-form reports is provided in Appendix B.2.

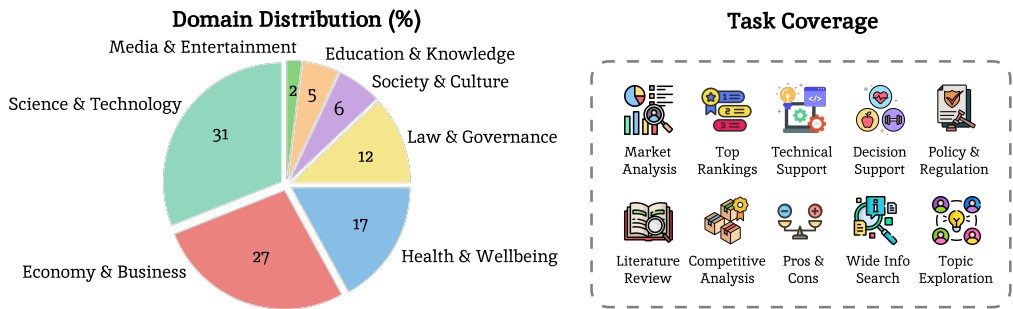

Figure 2: Domain distribution and task coverage of LiveResearchBench.

# 3 LIVERESEARCHBENCH

In this section, we present LiveResearchBench, our benchmark for evaluating agentic systems on challenging deep research tasks. We begin by outlining the key principles that govern task design (Section 3.1), then provide a high-level overview of the benchmark itself (Section 3.2). We then describe the data generation pipeline (Section 3.3) and conclude with our verification procedure to ensure benchmark quality (Section 3.4).

## 3.1 TASK DESIGN PRINCIPLES

To inform the design of our deep research benchmark, we conducted a user survey with participants from diverse backgrounds and occupations, including enterprise professionals, academic researchers and students, and users performing everyday tasks (see Appendix I). The survey aimed to identify realistic and representative use cases of deep research systems across varied contexts. From this study, we distilled four key principles to guide task design:

- **User-centric**: Tasks should reflect the needs of the *intended audience*. For example, a query posed by researchers may require technical jargon and fine-grained detail, while such complexity could overwhelm non-expert users.
- **Unambiguous**: Tasks should be *clearly specified*, since the quality of the final output depends heavily on how well user requirements are interpreted.
- **Time-varying**: Task requiring *real-time search* represent a large share of user queries. Such tasks are inherently resistant to data contamination from pretraining corpora, unlike static or time-bounded queries that risk becoming outdated or leaked as LLMs evolve.
- **Multi-faceted and search-intensive**: Tasks should be sufficiently challenging, demanding multi-hop search across *diverse web sources and in-depth analysis*. This ensures evaluation of deep research systems extends beyond simple fact retrieval.

## 3.2 BENCHMARK OVERVIEW

LiveResearchBench is designed as a user-centric, unambiguous benchmark for deep research tasks that require broad, real-time information gathering. It consists of 100 expert-curated questions, each paired with detailed checklists, created through a six-stage curation pipeline (Figure 3) and validated via a five-step quality control process (Figure 4). It spans seven domains—Science & Technology, Economy & Business, Health & Wellbeing, Law & Governance, Society & Culture, Education & Knowledge, and Media & Entertainment—and ten task categories, including market analysis, technical support, decision support, policy and regulation, literature review, competitive analysis, pros-and-cons comparison, wide information search, and topic exploration (Figure 2). Example queries is shown in Figure 1, with additional examples in Appendix F.

Compared to prior deep research benchmarks, LiveResearchBench tasks are designed to be more unambiguous and realistic (Figure 1). For example, Deep Research Bench (Bosse et al., 2025) includes search-intensive queries such as `How many IM and GM account closures did chess.com report for 2024?`, but these are static, time-bounded, and require minimal

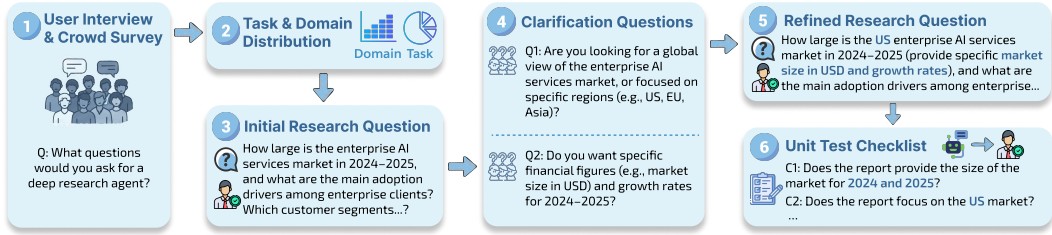

Figure 3: Six-stage data generation pipeline for LiveResearchBench. We begin with user interviews and surveys to capture realistic research needs, followed by expert-drafted research questions aligned with the identified domains. Clarification questions from frontier LLMs help ensure unambiguous scope, which human experts then refine into finalized queries. Finally, GPT-5 generates checklists that decompose each query into verifiable unit tests, enabling consistent coverage evaluation across systems. The quality of these checklists is further validated by human experts (Figure 4).

reasoning—limiting their ability to evaluate evolving real-world research. Likewise, DeepResearch Bench (Du et al., 2025) often omits crucial elements such as the target audience, content and format requirements, or a clearly defined scope. Such omissions leave queries open to multiple interpretations, even across runs of the same model, undermining evaluation reliability.

In contrast, tasks in LiveResearchBench are explicitly multi-faceted and grounded in user needs. For instance, the query in Figure 1 `Create a comprehensive report about the evolution of artistic styles ...Cover major historical periods from Ancient civilizations up to the present {{date}}. ...Target the report for undergraduate-level students ...Structure as a formal academic report ...Include analysis of major artists, techniques, materials, and stylistic characteristics for each period and region.` requires dynamic, up-to-date information search (with {{date}} replaced by the evaluation date), while also specifying the breadth and depth of coverage, the intended audience (undergraduate students), and the desired output format (a formal academic report). This level of clarity ensures consistent interpretation across models and imposes rigorous demands on information retrieval and in-depth reasoning.

## 3.3 BENCHMARK CONSTRUCTION

We curate the queries and checklists in LiveResearchBench through a six-stage pipeline (Figure 3). To capture realistic research needs, we start with user interviews and a crowd survey involving enterprise professionals, academics, and the general public. Participants were asked: `What questions would you ask a deep research agent?` (see Appendix I). The collected responses are then used to determine the task and domain distribution. Next, we hire domain experts from enterprise and academia to draft initial research questions aligned with these distributions.

To make the questions unambiguous and consistently interpretable across both models and human evaluators, we then prompt two top-tier deep research models, OpenAI o3 Deep Research and Gemini Deep Research, to generate possible clarification questions. Relying on a single model risks bias or omission, so using multiple models provides complementary perspectives. Human experts then review these clarifications, combine them with their domain expertise, and refine each query to establish a clear research scope and ensure a shared understanding of intent.

After the queries are finalized, we use GPT-5 to generate initial checklists. Each checklist decomposes a query into unit questions that test specific aspects of the required output. For example, for the query `How large is the US enterprise AI services market in 2024-2025 (provide specific market size in USD and growth rates)?...`, the corresponding checklist items include: 1) `Does the report provide the market size for 2024 and 2025?` 2) `Does the report focus on the US market?...` (9 checklist items in total). These checklists function as *unit tests* for the *coverage* dimension, verifying whether generated reports address all essential points in the query. This design enables consistent, reliable evaluation across different agentic systems.

## 3.4 DATA VERIFICATION

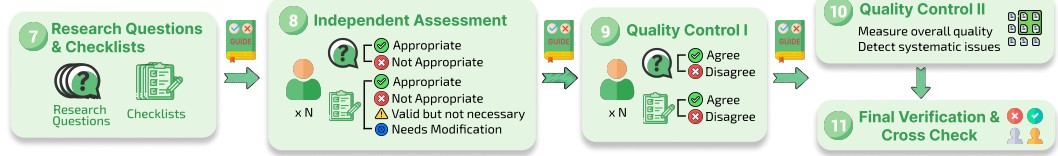

Figure 4: Five-stage data verification pipeline. Research questions and checklists are first independently assessed by expert annotators under detailed guidelines. Two rounds of quality control then ensure agreement across annotators and detect systematic issues. Finally, a separate group of experts conducts verification and cross-checking to resolve conflicts and finalize the dataset.

A rigorous verification process by human experts is essential for benchmark construction. We adopt a five-stage verification pipeline (Figure 4). After collecting the research questions and corresponding checklists, we prepare detailed annotation guidelines for expert annotators (Appendix J). Following these guidelines, annotators independently assess each query and checklist item. Queries are labeled as either *appropriate* or *not appropriate*, while checklist items are labeled as *appropriate*, *not appropriate*, *valid but not necessary*, or *needs modification*. Next, a separate group of quality-control experts reviews these annotations and independently decides whether they agree or disagree with the initial assessments. A second quality-control stage then samples annotation results to detect potential systematic issues and to ensure that overall quality deviates by no more than 5% from the true data quality. Finally, a third group of experts performs the last round of verification and cross-checking. They resolve conflicts, refine the queries and checklists as needed, and finalize the dataset.

## 4 DEEPEVAL: A COMPREHENSIVE EVALUATION SUITE FOR DEEP RESEARCH

Evaluating long-form, open-ended generation poses unique challenges: responses are too diverse for string matching, real-time queries lack fixed ground truths, and tasks are inherently multi-faceted. To address this, we introduce DeepEval, a comprehensive evaluation suite for model-generated research reports. DeepEval assesses both report- and content-level quality across six complementary dimensions: ❶ Presentation & Organization, ❷ Factual & Logical Consistency, ❸ Coverage & Comprehensiveness, ❹ Analysis Depth, ❺ Citation Association, and ❻ Citation Accuracy.

**Evaluation Protocol.** As the metrics differ in granularity, no single evaluation protocol is sufficient. For metrics ❶-❺[1], we consider three options (Gu et al., 2025): (1) `Checklist-based`, where the judge evaluates the generated report against a checklist and provides a binary score (0/1) for each item in the checklist. (2) `Pointwise (additive)`, where the judge aims to find concrete errors in the report according to the criteria. The final score is determined based on the number of errors. (3) `Pairwise comparison`, where the judge compares two reports side by side and selects the preferred one (or declares a tie). Based on preliminary experiments, we narrowed down the choice of protocol for each metric, and subsequently conducted a human alignment study on the selected protocols. As a result, we adopt `Checklist-based` for ❶ Presentation & Organization and ❸ Coverage & Comprehensiveness; `Pointwise (additive)` for ❷ Factual & Logical Consistency and ❺ Citation Association; `Pairwise comparison` for ❹ Analysis Depth. These protocols achieve high agreement with human judgments (see Appendix C).

*Remarks:* Another possible evaluation protocol is to prompt an LLM judge to assign a single rating within a fixed range (*e.g.*, 0-10), guided by rubrics that define the meaning of each score. However, our initial study showed that this approach is unsuitable for any of our metrics. Even with SoTA judges (Gemini-2.5 Pro and GPT-5), agreement with human judgments fell below 60%, and the assigned scores exhibited large variability across runs, for example, differences exceeding 50 points when directly rating analysis depth. Upon closer examination of the judge responses, we found that the judge models often produced conflated scores with shallow reasoning (e.g., "the report is overall well written because..."), rather than systematically identifying issues sentence by sentence. In other words, once a model commits to an overall positive assessment, it rarely examines the report in detail, leading to missed errors and inflated scores.

---

[1]❻ Citation Accuracy requires verifying (statement, URL) pairs throughout the report. This task is naturally handled using the rubric tree protocol detailed in Section 4.2.

**Agent-Ensemble-as-a-Judge.** To mitigate inductive bias from relying on a single model, we conducted a pilot study to identify which top-tier models align best with human preferences. Among the candidates—Gemini 2.5 Pro, GPT-5, and Claude 4 Sonnet—results showed that Gemini 2.5 Pro aligned most closely, followed by GPT-5, while Claude 4 Sonnet exhibited inconsistent and low agreement with human experts (Appendix C). Based on these findings, we adopt a multi-judge ensemble protocol for all LLM-as-a-Judge evaluations. Specifically, we use Gemini 2.5 Pro and GPT-5 as independent judges and report final scores as the average of their assessments.

### 4.1 Coarse-grained: Report-level Metrics

❶ **Presentation & Organization.** Poorly organized reports or those with frequent grammatical errors undermine user trust. We evaluate presentation quality with a checklist of 10 common error patterns from our pilot study (Table 2), covering structure, grammar, citations, duplicates, and formatting. Each item is scored in binary form $(0/1)$, and we report the average success rate across items and an ensemble of judge models as this configuration best aligns with human judgment—achieving agreement in 98.3% of cases (Appendix C). Examples of checklist items include: *(1) Does the report contain any grammar or spelling errors? (2) If figures or tables are included, do they present complete data or valid visual elements? (3) Does every reference entry correspond to at least one in-text citation?* Full checklists and prompts are provided in Appendix A.1.

❷ **Factual & Logical Consistency.** As reports grow longer, they are more likely to contain factual and logical inconsistencies, or even contradictory claims and numbers. This metric captures whether claims remain coherent and factually consistent throughout the document. We adopt the pointwise (additive) evaluation protocol based on the human alignment study, where the LLM judge is tasked with identifying as many substantive inconsistencies as possible. The final score is then computed mechanistically based on the number of issues detected, scaled from 0 to 100: a report with no contradictions receives a score of 100, while each additional inconsistency leads to a deduction. Detailed prompts are provided in Appendix A.2.

### 4.2 Fine-grained: Content-level Metrics

❸ **Coverage & Comprehensiveness.** A strong report should directly address all aspects of the multi-faceted query. To make evaluation consistent and interpretable, we leverage the human curated and verified checklists introduced in Section 3. Each checklist item functions as a unit test: the judge assigns a binary score (0/1) indicating whether the requirement is met, and we report the average success rate across all items and queries. The judge-model ensemble aligns with human judgment—100.0% of preferences favor either or both. The evaluation prompt is provided in Appendix A.3.

❹ **Analysis Depth.** Beyond coverage, a strong research report should provide substantive insights rather than superficial information gathering. We assess depth along five dimensions: (1) *granularity of reasoning* (causal chains), (2) *multi-layer insights* (trade-offs and implications), (3) *critical evaluation* (critiques and limitations), (4) *analytical use of evidence*, and (5) *insight density*. Ensemble judges conduct pairwise comparisons between the evaluated and baseline reports, assigning 1–5 scores to each along every dimension. To mitigate positional bias, we adopt the position-swap averaging method of Wang et al. (2023), scoring each pair twice with reversed order and averaging results. Dimension scores are summed, averaged across judges, and used to derive win rates: A win, B win, or Tie (if scores differ by $\leq 1$). Ties are excluded from the denominator. The human agreement rate is 92.5% with the judge ensemble. Detailed prompts can be seen in Appendix A.4.

❺ **Citation Traceability.** This metric checks whether all factual claims are linked to corresponding sources. For example, a statement like "medium-lift launch vehicles held 56.63% of the market in 2024" should include proper source URL. The LLM judge aims to flag claims not properly cited, and compute a score scaled 0–100: full coverage yields 100, with deductions for missing or misplaced citations, similar to the scoring mechanism in ❷. The human agreement rate is 85.9% with the judge ensemble. Detailed prompts can be seen in Appendix A.5.

❻ **Citation Accuracy.** Complementary to association, this metric evaluates whether citations genuinely support their claims. In citation-grounded reports, links may be hallucinated, irrelevant to the topic, or fail to substantiate the associated statement. To capture this, we employ an agentic judge with web access that performs a structured validation process: (1) identify each claim with its linked source URL, (2) check whether each URL is accessible, (3) if accessible, assess whether the content sufficiently supports the claim. This procedure is organized as a rubric tree in Figure 13.

## 5 MAIN RESULTS AND ANALYSIS

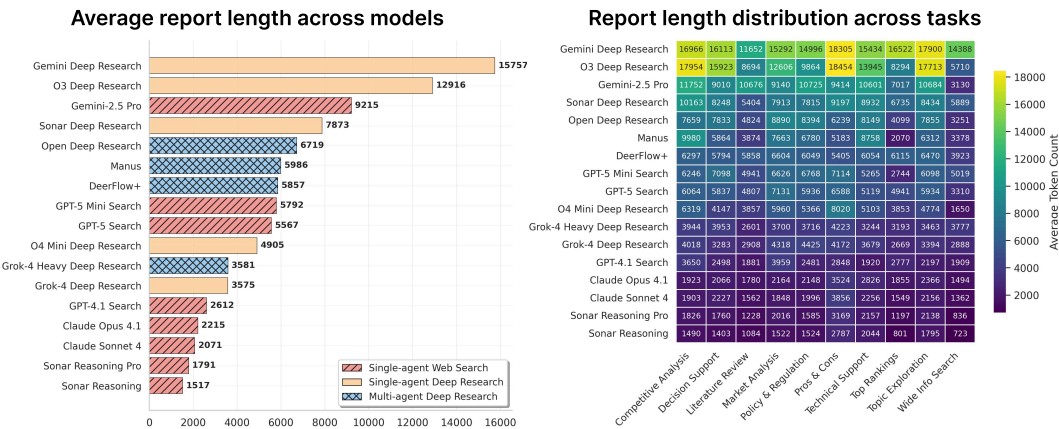

Figure 5: Distribution of report lengths across systems and tasks.

**Agentic Systems.** We conduct a comprehensive evaluation of 17 state-of-the-art open-sourced and proprietary agentic systems, which can usually be grouped into three categories: (1) *Single-agent systems with web search capabilities*, including GPT-5 (OpenAI, 2025a) , GPT-4.1 (OpenAI, 2024), GPT-5-mini (OpenAI, 2025b), Gemini 2.5 Pro (DeepMind, 2025b), Gemini 2.5 Flash (DeepMind, 2025a), Claude 4 Sonnet (Anthropic, 2025a), Claude 4.1 Opus (Anthropic, 2025b), Perplexity Sonar Reasoning (Perplexity, 2025a), and Perplexity Sonar Reasoning Pro (Perplexity, 2025b); (2) *Single-agent deep research systems*, which feature extended reasoning depth and longer thinking time, including OpenAI o3 Deep Research (OpenAI, 2025c), OpenAI o4-mini Deep Research (OpenAI, 2025d), Perplexity Sonar Deep Research (AI, 2025b), Grok-4 Deep Research (Expert) (xAI, 2025b), and Gemini Deep Research (DeepMind, 2025c); (3) *Multi-agent deep research systems*, which coordinate a team of specialized agents to decompose complex queries. These systems typically employ a planner to dispatch sub-agents for browsing, searching, synthesis, and verification, while maintaining a large shared context across tasks. This category includes Manus (Manus, 2025), Grok-4 Heavy Deep Research (xAI, 2025a), Open Deep Research (AI, 2025a), and Deerflow+. Deerflow+ is our enhanced implementation of Deerflow (ByteDance, 2025) that resolves frequent failure modes and early termination by adding inline citations, reference mapping, and robust long-context management. Implementation details are provided in Appendix D; Figure 12 shows side-by-side outputs for the same prompt in Figure 1.

*Remarks:* Non-reasoning models and multi-agent systems that do not excel at generating long-form reports are excluded due to poor performance in our pilot study. For each system, we configure settings to maximize performance. For example, setting output length to the maximum token limit (128,000 for GPT-5 models) and enabling retries for low-quality or failed generations. For open-source systems like Deerflow+ and Open Deep Research, we use GPT-5 as the backbone model.

**Obs.❶ Longer reports are common, but not consistently better.** Figure 5 shows the distribution of report lengths across models and tasks. Single-agent deep research systems backed by strong base LLMs (Gemini Deep Research, OpenAI o3 Deep Research) produce substantially longer reports than both multi-agent systems and single-agent systems with web search. Yet, multi-agent pipelines do not automatically yield longer reports. For example, Deerflow+ and Open Deep Research rely on the final synthesizer to produce the report given searched findings, which constrains length. We also observe task-level variation on report length is smaller than model-level variation. Length can also be inflated by *formatting choices* rather than substance. Some systems emit very long, provider-specific redirect URLs in citations (e.g., "grounding" redirect URLs for Gemini 2.5 Pro), or duplicate links both inline and in the bibliography (OpenAI o3 Deep Research), effectively counting the same URL twice. Since we use shorten links and cite once for multi-agent systems for Deerflow+ and Open Deep Research, they yield fewer URL tokens. Hence, token counts reflect citation handling as much as content. As we show later, longer reports do not necessarily translate into higher quality; improvements in citation discipline and factual consistency are not monotonically related to length.

| Agent Name | Presentation & Organization | Fact & Logic Consistency | Coverage & Comprehensiveness | Citation Association |
|---|---|---|---|---|
| **Single-Agent with Web Search** | | | | |
| GPT-5 | 71.6 | 68.3 | 83.4 | 67.6 |
| GPT-4.1 | 66.0 | 65.9 | 63.6 | 50.6 |
| GPT-5-mini | 61.4 | 66.9 | 80.5 | 55.5 |
| Gemini 2.5 Pro | 51.9 | **76.5** | 73.1 | 38.5 |
| Claude 4 Sonnet | 81.9 | 67.3 | 49.2 | 37.9 |
| Claude 4.1 Opus | 81.6 | 67.5 | 50.8 | 35.6 |
| Perplexity Sonar Reasoning | 82.1 | 73.0 | 40.7 | 52.6 |
| Perplexity Sonar Reasoning Pro | 79.6 | 71.9 | 46.7 | 50.1 |
| **Single-Agent Deep Research** | | | | |
| OpenAI o3 Deep Research | 71.3 | 64.2 | 85.0 | 25.6 |
| OpenAI o4-mini Deep Research | 74.3 | 62.3 | 78.6 | 27.2 |
| Perplexity Sonar Deep Research | **83.5** | 67.4 | 65.5 | 36.6 |
| Grok-4 Deep Research | 69.1 | 57.4 | 86.3 | 49.5 |
| Gemini Deep Research | 62.1 | 63.0 | 75.8 | 52.1 |
| **Multi-Agent Deep Research** | | | | |
| Manus | 75.0 | 63.1 | 73.3 | 45.6 |
| Grok-4 Heavy Deep Research | 75.9 | 59.4 | **89.3** | 48.0 |
| Deerflow+ (w. GPT-5) | 78.8 | 69.9 | 61.6 | **77.0** |
| Open Deep Research (w. GPT-5) | 81.0 | 71.3 | 65.3 | 76.9 |

Table 1: Evaluation of single-agent and multi-agent deep research systems across four dimensions. Scores (0–100) are normalized evaluation outcomes averaged across all tasks and LLM judge ensembles. We **bold** the best across three categories and underline the best agent for each category.

**Obs.❷ Models struggle most with citation correctness and formatting, rather than surface fluency.** Across systems, we repeatedly observe failure modes that are trivial for humans but challenging for current agents, such as mismatched in-text citations and references (Figure 14), missing or incomplete URLs (Figure 15), inconsistent citation formats (Figure 16), uncited references appearing in the bibliography (Figure 17), broken or incomplete table formatting (Figure 18), and out-of-order reference numbering (Figure 19). These recurring issues motivated the design of DeepEval's Citation Association dimension (Section 4), and foreshadow the quantitative gaps we analyze next.

Table 1 summarizes performance across four DeepEval dimensions. Below we highlight key insights.

**Obs.❸ Multi-agent families lead on average.** At the system level, Open Deep Research achieves the highest average (73.6), followed by GPT-5 (72.7) and Deerflow+. Averaged by family, multi-agent deep research systems (69.5) outperform both single-agent web (62.8) and single-agent deep research, indicating that collaborative decomposition and aggregation provide reliable gains.

**Obs.❹ Single web agent excels in consistency, multi-agent systems in association, while single-agent deep research lags.** Single-agent web models are strongest on Factual & Logical Consistency (69.7 avg), benefiting from a single persistent memory stream that maintains reasoning continuity. Gemini 2.5 Pro achieves the highest consistency score overall (76.5), with Perplexity Sonar Reasoning and Perplexity Sonar Reasoning Pro close behind (73.0 and 71.9). For example, single web agents such as Gemini 2.5 Pro maintain coherence because the absence of inter-agent handoffs reduces the risk of logical drift. Similarly, Perplexity Sonar Reasoning benefits from producing relatively concise reports, which lowers the probability of factual or logical mistakes while keeping reasoning aligned. Multi-agent systems dominate Citation Association (61.9 avg), where explicit steps for aligning sub-agent outputs with citations or dedicated citation agents yield cleaner claim–evidence links. By contrast, single-agent deep research models perform worst on Citation Association, primarily because many key statements remain uncited and URLs are often mismatched. For example, OpenAI o3 Deep Research often produces long reports but leaves key statements/facts uncited, while Grok-4 Heavy Deep Research often cites URLs that fail to support the associated claims. Another common recurring issue is the use of fictional or non-existent links, as seen in Manus.

**Obs.❺ Multi-agent systems lead presentation, but surface polish does not imply grounded quality.** Multi-agent systems achieve the highest presentation scores (77.7 avg.), with Open Deep Research (81.0) and Deerflow+ (78.8) producing the most polished outputs. Yet, presentation is weakly correlated with association or consistency—visually well-organized reports may still contain inconsistent claims.

**Obs.❺ Coverage benefits from specialization, but scaling retrieval scope and system complexity strain memory capacity.** On average, single agents like GPT-5 series and the Grok family perform strongly on Coverage & Comprehensiveness, with Grok-4 Heavy Deep Research achieving the highest score (89.3). By contrast, Deerflow+ (61.6) and Open Deep Research (65.3) lag despite strong citation association. The bottleneck arises as the search context scales: even hundreds of retrieved webpages can quickly exceed the available context window. In multi-agent systems, increasing planning steps and subagents can easily push retrieval into thousands (>3,000), where the problem compounds. Without robust mechanisms to label related content, track what has already been processed, and compress information without discarding key evidence, systems lose critical coverage. Thus, the open challenge is not the synthesizer alone, but the design of system-wide memory architectures and compression strategies that allow agents to retain breadth while maintaining fidelity.

**Obs.❼ Most systems are deep searchers, not deep researchers.** Figure 6 benchmarks analysis depth as win rate over Open Deep Research (with GPT-5 as backbone). Only Deerflow+ and Gemini Deep Research exceed ODR. Models that generate very long reports (e.g., OpenAI o3 Deep Research) often fail to synthesize across sources; Grok-4 Heavy Deep Research, though best on Coverage & Comprehensiveness, barely outperforms ODR in depth. This suggests that many systems behave more like *information collectors and organizers* than writers of evidence-grounded, argumentative reports—functioning as deep searchers rather than deep researchers that provide in-depth analysis. We provide further analysis with human judgment in Appendix C.

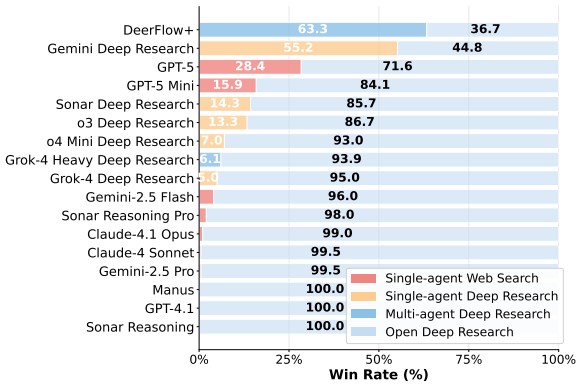

Figure 6: Analysis depth (win rate over Open Deep Research). Most systems collect and organize information but struggle to synthesize deeper insights.

**Obs.❽ Even SoTA systems are far from citation error-free.** Using our rubric-tree framework, we evaluate the top performers from Table 1 (GPT-5, Grok-4 Deep Research, and Open Deep Research) on the two most search-intensive tasks: market analysis and wide information search. The results are shown in Table 7, where for each agentic system and task category, we report the average number of E1, E2, and E3 errors in the generated reports. We can see that all models produce non-trivial citation errors. For example, in wide information search, most errors stem from *unsupported claims* (claims not verifiable from the cited link) rather than invalid or irrelevant URLs, highlighting that hallucinations persist even with web access. Further details are provided in Appendix E.

**Obs.❾ Coherence, verifiability, breadth, depth trade off under context limits.** Single web agent systems favor coherent reasoning; multi-agent systems achieve stronger association and presentation; and specialized designs lead in coverage. Yet no system achieves all four. Sustaining both high coverage and depth will require: (i) *long-horizon memory with update operations*, (ii) *hierarchical information compression without information loss*, and (iii) *explicit synthesis/argumentation modules*. Effective compression should address two challenges: (1) when information is partially redundant, the system should merge overlapping content without discarding unique evidence; and (2) when all information is related, compression should instead operate by assigning *importance levels* to retain critical details within the context limit. Because different users may emphasize different aspects of information, such representations must be both *importance- and preference-aware*, capturing what is essential for reasoning or verification while adapting to user priorities. Without these capabilities, scaling retrieval primarily increases report length rather than reliable, user-aligned insight.

## 6 CONCLUSION

We introduced LiveResearchBench, a benchmark of 100 expert-curated, dynamic tasks for deep research, and DeepEval, a comprehensive evaluation suite spanning six dimension for citation-grounded reports. Our evaluation of 17 leading systems shows that while agents can gather and organize information, they struggle with citation reliability and analytical depth. LiveResearchBench and DeepEval establish a rigorous foundation for benchmarking and point toward future advances in memory, compression, and synthesis as key to enabling truly insightful deep research.

ACKNOWLEDGEMENT

The authors would like to thank ICLR anonymous reviewers for their insightful feedback and helpful discussions. This project is funded by Salesforce. We are also grateful for the support of the National Science Foundation (NSF) (CCF2106707, CCF2446711), the Defense Advanced Research Projects Agency (DARPA Young Faculty Award), and the Wisconsin Alumni Research Foundation (WARF).

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

# Appendix

## A EVALUATION DETAILS

### A.1 PRESENTATION & ORGANIZATION

**Grading prompt and checklist.** The prompt for presentation and organization is shown in Figure 7, and the corresponding checklist questions are provided in Table 2. The same checklist is used for all research questions.

---

**Evaluation Prompt for Presentation & Organization (Checklist-based)**

<system_role>
You are an expert evaluator assessing research reports for presentation quality and formatting standards. Your sole task is to determine if the report meets specific presentation criteria with binary scoring (0 or 1).

**EVALUATION CRITERIA:**
- Score 1: The report fully satisfies the presentation criterion with no issues.
- Score 0: The report fails to meet the presentation criterion or has any issues that prevent it from passing.

**INSTRUCTIONS:**
- Read the research report carefully to assess presentation quality, formatting, and structural elements.
- For each presentation criterion, determine if the report fully meets the standard.
- Provide a binary score (0 or 1) for each criterion.
- Provide a clear justification for your score, referencing specific elements in the report.

**IMPORTANT GUIDELINES:**
- Strict Binary Assessment: A score of 1 requires complete satisfaction of the criterion. Any failure to meet the standard results in a score of 0.
- Focus on Presentation: Evaluate only presentation quality, formatting, structure, grammar, citations, and formatting consistency. Do not evaluate content accuracy, research quality, or factual correctness.
- Citation Standards: For citation-related criteria, carefully check that all in-text citations have corresponding reference entries and vice versa.
- Grammar and Spelling: For grammar/spelling criteria, any errors result in a score of 0.
- Formatting Consistency: Check for consistent use of formatting elements like headers, citation styles, etc.
  Respond with a JSON object containing your evaluations for each presentation criterion.
</system_role>
<user_prompt>
ORIGINAL RESEARCH QUERY:
{query}
PRESENTATION CHECKLIST ITEMS TO EVALUATE:
{checklist_section}
RESEARCH REPORT TO EVALUATE:
{report_content}
</user_prompt>

---

Figure 7: Instruction for evaluating **presentation and organization quality** of research reports. Placeholders such as {user_prompt}, {query}, {checklist_section} and {report_content} are filled at runtime.

### A.2 FACTUAL & LOGICAL CONSISTENCY

**Grading prompts and rubrics.** The grading prompt for factual and logical inconsistency is provided in Figure 8. The scoring rubrics are provided in Table 3. The scores for Factual & Logical Consistency are inversely correlated with the number of detected inconsistencies. Users can adjust the penalty weight assigned to each error depending on the evaluation need.

---

**Evaluation Prompt for Factual and Logical Consistency (Pointwise Additive)**

<system_role>
You are an expert evaluator assessing research reports based on the Criterion Description: Factual and Logical Consistency.

**INSTRUCTIONS:**

---

| No. | Checklist Questions for Report Presentation Quality |
|-----|------------------------------------------------------|
| 1 | Does the report present a clear, coherent, and logically ordered structure so that the organization is easy to follow and directly addresses the research question? |
| 2 | Does the report contain zero grammar and spelling errors? |
| 3 | Does every entry in the reference list correspond to at least one in-text citation? |
| 4 | Does every in-text citation have a corresponding entry in the reference list? |
| 5 | Is there exactly one "References" (or "Bibliography" / "Sources") section, and are its entries sorted according to a single, consistent scheme? |
| 6 | Is a single, consistent citation style used throughout the entire document? |
| 7 | Are all in-text citations placed logically at the end of a clause or sentence, without interrupting grammatical flow? |
| 8 | If the report includes figures or tables, does each one contain complete data or a valid visual element? (If none are included, the report automatically passes this test.) |
| 9 | Is the formatting correct and consistent? For example: (a) If delivered in Markdown, are proper heading levels (#, ##, etc.) used instead of plain text for section titles; (b) if Markdown tables are included, is their syntax valid and renderable? |
| 10 | If the citations are numbered, are there no skipped numbers (e.g., [23], [25], [26] with [24] missing) and no duplicates (two different sources assigned the same number, or one source assigned multiple numbers)? |

Table 2: Checklist questions for evaluating report **Presentation & Organization** quality.

- Be very careful and detail-oriented. Read the report sentence by sentence and identify concrete issues.
- Be critical and thorough in your evaluation: do not overlook problems or give inflated scores.
- Find as many substantive issues **relevant to the criterion** as possible.
- Issues must be genuine violations that clearly dissatisfy the criterion, not minor nitpicks or subjective preferences.
- Do not include issues that are irrelevant to the criterion.

**Criterion Description: Factual and Logical Consistency**
**Score rubrics:**
{Score rubrics table}

**Focus on:**
- Logical inconsistencies
- Factual contradictions (facts, claims, numbers, dates, names, etc.)

**Remarks:**
1. Factual accuracy (e.g., whether the report is free of factual errors) is a separate criterion and should NOT be considered here.
2. The same source can be used to cite multiple claims. This does NOT constitute a contradiction.

**OUTPUT FORMAT:**
Respond in JSON with the following fields:
- `specific_issues`: A list of specific problems, with exact quotes or locations in the text (e.g., "In section X...", "The claim that '...' is unsupported").
- `total_issues`: Total number of issues (must match the length of `specific_issues`).
- `score`: An integer from 1–10 based on the rubric and issues identified.
- `reasoning`: A detailed explanation (2–3 sentences) summarizing your assessment and referencing the identified issues.
</system_role>
<user_prompt>
**TASK:**
{task}

**REPORT TO EVALUATE:**
{report}

Please evaluate this report on the criterion and provide your assessment in JSON format.
</user_prompt>

Figure 8: Instruction for evaluating research reports on logical and factual inconsistency. The rubrics table is provided in Table 3. Placeholders such as {task} and {report} are dynamically filled at runtime.

| Score | Issue Count | Description |
|-------|-------------|-------------|
| 100 | 0 | Perfect – No contradictions or inconsistencies detected |
| 90 | 1–2 | Excellent – Very minor inconsistencies |
| 80 | 3–4 | Very Good – Few minor inconsistencies |
| 70 | 5–6 | Good – Some inconsistencies |
| 60 | 7–8 | Above Average – Several inconsistencies |
| 50 | 9–10 | Average – Multiple inconsistencies |
| 40 | 11–12 | Below Average – Many inconsistencies |
| 30 | 13–14 | Poor – Extensive inconsistencies |
| 20 | 15–17 | Very Poor – Pervasive inconsistencies |
| 10 | 18+ | Unacceptable – Overwhelming inconsistencies |

Table 3: Scoring rubric for evaluating inconsistencies.

## A.3 COVERAGE & COMPREHENSIVENESS

**Grading prompt and checklist.** We present the prompt for evaluating Coverage & Comprehensiveness in Figure 9. An example coverage checklist is provided in Table 4. Note that the checklist for every research question is different (Sections 3.3 and 3.4).

---

**Evaluation Prompt for Coverage & Comprehensiveness (Checklist-based)**

<system_role>
You are an expert evaluator assessing research reports against specific checklist criteria derived from the original research query.
Your sole task is to determine if the report fully and completely delivers the specific data and information requested for each item.
EVALUATION CRITERIA:
- Score 1: The report fully and completely provides all specific data and information required by the checklist item.
- Score 0: The report fails to provide the required data or provides an incomplete response.
INSTRUCTIONS:
- Read the original research query to understand the exact data being requested.
- Read the research report to find the data that directly and completely answers the query.
- For each checklist item, determine if the report delivers all required components of the answer.
- Provide a binary score (0 or 1).
- Provide a clear justification for your score, referencing the completeness or incompleteness of the provided data.
IMPORTANT GUIDELINES:
- Strict Requirement for Completeness: A score of 1 requires 100% fulfillment of the checklist item. If any part of the requested information for that item is missing, incorrect, or incomplete, the score must be 0. For example, if a checklist item requires a name, a date, and a valid link, the report must provide all three to receive a score of 1. Providing only two of the three results in a score of 0.
- Data vs. Methodology: Your evaluation must distinguish between a report that provides the answer versus one that describes a process to find the answer. A methodology, plan, or description of how the data could be found is not a substitute for the data itself and must be scored 0.
- No Credit for Placeholders: A report that acknowledges a checklist item but explicitly states the information is missing, unavailable, or not provided (e.g., "Information not provided," "Data not found") must receive a score of 0 for that item.
- Focus on Delivery: Base your evaluation strictly on the final data delivered in the report. Do not score based on what the report promises, implies, or outlines as its goal.
Respond with a JSON object containing your evaluations for each checklist item.
</system_role>
<user_prompt>
ORIGINAL RESEARCH QUERY:
{query}
CHECKLIST ITEMS TO EVALUATE:

---

```
{checklist_section}
RESEARCH REPORT TO EVALUATE:
{report_content}
</user_prompt>
```

Figure 9: Instruction for evaluating Coverage & Comprehensiveness of the generated reports. Placeholders `{query}`, `{checklist_section}` and `{report_content}` are filled at runtime.

| ID | Sample Checklist Questions |
|---|---|
| 1 | Does the report cover major historical periods from Ancient civilizations up to the present (evaluation date)? |
| 2 | Does the report focus primarily on visual arts, specifically painting, sculpture, and architecture? |
| 3 | Does the report provide analysis of key regional or national characteristics for European, East Asian (including China and Japan), Islamic, and Indigenous American art? |
| 4 | Is the report structured as a formal academic report, including proper citations and references? |
| 5 | Does the report analyze how political, social, religious, and technological factors influenced artistic development? |
| 6 | Does the report include analysis of major artists, techniques, materials, and stylistic characteristics for each period and region? |

Table 4: Checklist questions for evaluating Coverage & Comprehensiveness, corresponding to the example in Figure 1.

## A.4 ANALYSIS DEPTH

**Grading prompt.** The prompt for Analysis Depth is shown in Figure 10. We use pairwise comparison, as detailed in Section 4.2.

---

**Evalaution Prompt for Analysis Depth (Pairwise)**

<system_role>
You are an impartial research report evaluation expert. Your task is to compare **Report A** and **Report B** side-by-side and decide which demonstrates **greater depth of analysis**.

**What "Depth of Analysis" Means**
Depth = *how far the report goes beyond surface description into reasoning, insight, and layered analysis.*
**Scoring Dimensions (0–5 each)**
Assign each report a score on these five dimensions:
1. **Granularity of Reasoning (0–5)**
- 0 = purely abstract or vague
- 3 = some unpacking into mechanisms or details
- 5 = consistently breaks down ideas into specific causal chains or subcomponents
2. **Multi-Layered Insight (0–5)**
- 0 = only surface-level restatement
- 3 = includes some implications or second-order effects
- 5 = consistently explores trade-offs, deeper implications, "so what" insights
3. **Critical Evaluation (0–5)**
- 0 = passive description only
- 3 = some questioning of assumptions or limited contrast of alternatives
- 5 = strong critique, weighing of scenarios, probing of limitations
4. **Analytical Use of Evidence (0–5)**
- 0 = mentions facts/examples without connecting them to reasoning
- 3 = some evidence linked to argument
- 5 = evidence consistently advances or sharpens analysis

---

5. **Insight Density (0–5)**
- 0 = mostly filler or generic phrasing
- 3 = mix of insight and filler
- 5 = highly concentrated with substantive analysis per token

**Total Depth Score** - Sum the five criteria → total score (0–25).

**Judging Rules**
1. Compare both reports **only on depth**, using the rubric above.
2. Give each report a **0–25 depth score**.
3. Decide the outcome:
- If one report's score is more than 1 point higher → it wins.
- If the difference is ≤ 1 point → call it a tie.

**Exclude Entirely**
- Coverage (breadth of topics).
- Factual correctness or consistency.
- Presentation, formatting, style, grammar.
- Citation traceability.
- Length alone (verbosity ≠ depth).

**Output Format (JSON only)**

```
{
  "winner": "A | B | tie",
  "scores": {
    "A": {
    "granularity": 0-5,
    "insight": 0-5,
    "critique": 0-5,
    "evidence": 0-5,
    "density": 0-5,
    "total": 0-25
    },
    "B": {
    "granularity": 0-5,
    "insight": 0-5,
    "critique": 0-5,
    "evidence": 0-5,
    "density": 0-5,
    "total": 0-25}
  },
  "justification": "less than 80 words explaining which report shows
  deeper reasoning and why, referencing specific differences
  in depth.",
  "major_flaws": {
    "A": ["notes on shallow reasoning or lack of insight if any"],
    "B": ["notes on shallow reasoning or lack of insight if any"]
  }
}
```

</system_role>
<user_prompt>
Please compare these two reports ONLY in terms of depth of analysis relative to the research problem and determine which demonstrates greater analytical depth. {query}
REPORT A:
{report_a_content}
REPORT B:
{report_b_content}
</user_prompt>

Figure 10: Instruction for evaluating analysis depth of the generated reports. Placeholders {query}, {report_a_content}, and {report_b_content} are filled at runtime.

## A.5 CITATION TRACEABILITY

**Grading prompts and rubrics.** We provide the grading prompt for citation traceability in Figure 11. The scoring rubrics are provided in Table 5. As the scores are determined by the number of claims not properly cited, users can adjust the penalty weight assigned to each error depending on the use case.

---

**Evaluation Prompt for Citation Traceability (Pointwise Additive)**

<system_role>
You are an expert evaluator assessing research reports based on the Criterion Description: Citation Traceability.

**INSTRUCTIONS:**
- Be very careful and detail-oriented. Read the report sentence by sentence and identify concrete issues.
- Be critical and thorough in your evaluation: do not overlook problems or give inflated scores.
- Find as many substantive issues **relevant to the criterion** as possible.
- Issues must be genuine violations that clearly dissatisfy the criterion, not minor nitpicks or subjective preferences.
- Do not include issues that are irrelevant to the criterion.

**Criterion Description: Citation Traceability**
**Score rubrics:**
{Score rubrics table}

**Focus on:**
- Proper association of claims with citations. Each factual claim that requires evidence should be accompanied by a corresponding citation. Flag cases where claims lack citations or where citations are clearly mismatched. For example, a healthcare URL attached to a statement on market share such as "medium-lift launch vehicles held 56.63% of the market in 2024".

**Remarks:**
1. A URL may not always be attached immediately after a claim; in some cases, it appears at the end of a paragraph and is intended to support the preceding claims within that paragraph.
2. Factual accuracy (e.g., whether the report is free of factual errors) is a separate criterion and should NOT be considered here.

**OUTPUT FORMAT:**
Respond in JSON with the following fields:
- `specific_issues`: A list of specific problems, with exact quotes or locations in the text (e.g.,"In section X...", "The claim that '...' is unsupported").
- `total_issues`: Total number of issues (must match the length of `specific_issues`).
- `score`: An integer from 1–10 based on the rubric and issues identified.
- `reasoning`: A detailed explanation (2–3 sentences) summarizing your assessment and referencing the identified issues.
</system_role>
<user_prompt>
**TASK:**
{task}

**REPORT TO EVALUATE:**
{report}

Please evaluate this report on the criterion and provide your assessment in JSON format.
</user_prompt>

---

Figure 11: Instruction for evaluating research reports on citation traceability. The rubrics table is provided in Table 5. Placeholders such as {task} and {report} are dynamically filled at runtime.

# B EXPANDED RELATED WORKS

This section provides an expanded comparison of LiveResearchBench with representative benchmarks and evaluation protocols for long-form open-ended reports on information seeking and deep research.

## B.1 BENCHMARK COMPARISON

Table 6 summarizes the key dimensions, including human-verified rubrics/answers, expert-curation, open-ended task design, domain coverage, dynamism, and evaluation methodology.

| Score | Uncited Claims | Description |
|---|---|---|
| 100 | 0 | Perfect – All major claims/facts have citations; fully traceable |
| 90 | 1–2 | Excellent – Very few claims lack citations; minimal impact |
| 80 | 3–4 | Good – Few claims lack citations; minor omissions |
| 70 | 5–6 | OK – Some claims lack citations |
| 60 | 7–8 | Above Average – Several claims lack citations |
| 50 | 9–10 | Average – Many claims lack citations; significant association issues |
| 40 | 11–12 | Below Average – Most claims lack citations |
| 30 | 13–14 | Poor – Extensive uncited claims; report poorly supported |
| 20 | 15–17 | Very Poor – Overwhelming lack of citations |
| 10 | 18+ | Unacceptable – Report largely untraceable |

Table 5: Scoring rubric for evaluating citation traceability.

| Benchmark | Human-verified | Expert-curated | Open-ended | Multi-domain | Dynamic Queries | Agent-ensemble as-a-judge |
|---|---|---|---|---|---|---|
| ScholarSearch (Zhou et al., 2025) | ✗ | ✗ | ✗ | ✓ | ✗ | ✗ |
| BrowseComp (Wei et al., 2025) | ✓ | ✗ | ✗ | ✓ | ✗ | ✗ |
| ResearcherBench (Xu et al., 2025b) | ✓ | ✓ | ✓ | ✗ | ✗ | ✗ |
| DeepScholar-Bench (Patel et al., 2025) | ✗ | ✗ | ✓ | ✗ | ✓ | ✗ |
| ReportBench (Li et al., 2025) | ✗ | ✗ | ✗ | ✓ | ✓ | ✗ |
| DeepResearch Bench (Du et al., 2025) | ✗ | ✓ | ✓ | ✗ | ✗ | ✗ |
| Mind2Web2 (Gou et al., 2025) | ✓ | ✓ | ✗ | ✓ | ✗ | ✗ |
| LiveDRBench (Java et al., 2025) | ✗ | ✗ | ✗ | ✓ | ✗ | ✗ |
| Deep Research Bench (Bosse et al., 2025) | ✓ | ✓ | ✗ | ✓ | ✗ | ✗ |
| DeepResearchGym (Coelho et al., 2025) | ✗ | ✗ | ✗ | ✓ | ✗ | ✗ |
| **LiveResearchBench (Ours)** | ✓ | ✓ | ✓ | ✓ | ✓ | ✓ |

Table 6: Comparison of deep research benchmarks across key dimensions.

As shown in Table 6, no existing benchmark satisfies all criteria simultaneously. In contrast, LiveResearchBench combines human-verified rubrics, expert-curated tasks, open-ended multi-domain queries, time-varying scenarios, and an Agent-ensemble-as-a-judge to ensure consistency and reliability.

## B.2 EVALUATION PROTOCOLS COMPARISON

We analyze how prior benchmarks evaluate long-form or open-ended research outputs and how LiveResearchBench addresses their limitations.

**DeepResearch Bench** (Du et al., 2025) evaluates model-generated research reports using a single LLM call (Gemini-2.5-Flash) to assign holistic scores across four dimensions: comprehensiveness, depth, instruction-following, and readability. These are then aggregated using LLM-predicted weights. However, our pilot study indicates that single-judge protocols exhibit high variance. Independent judging runs using frontier models (GPT-5, Gemini-2.5-Pro, Claude-4-Opus, Claude-4-Sonnet) frequently differ by over 50 points in analysis depth, severely limiting reproducibility. To address this, DeepEval employs an Agent-ensemble-as-a-judge combined with checklist-based, additive, pairwise, and rubric-tree evaluation protocols. We leverage LLMs with web search APIs enabled, allowing agents to retrieve external information during reasoning. These methods substantially reduce variance and provide finer-grained diagnostics across dimensions.

**ResearcherBench** (Xu et al., 2025b) evaluates long-form outputs using a single LLM judge to score coverage, faithfulness, and groundedness, which roughly correspond to our metrics of Coverage & Comprehensiveness, Citation Traceability, and Citation Accuracy. While valuable, this setup omits several key aspects necessary for evaluating open-ended research reports. However, DeepEval expands the evaluation space through three additional dimensions: 1) Presentation & Organization. Assesses a report's structural clarity, stylistic coherence, formatting hygiene, and citation organization. Poor presentation increases cognitive load and can obscure otherwise valid reasoning, directly affecting interpretability and traceability. 2) Analysis Depth. Evaluates a model's ability to perform nontrivial

analytical reasoning beyond surface-level retrieval or summarization. This guards against superficial optimization strategies that produce shallow but well-cited text. 3) Factual & Logical Consistency. Measures global coherence across the entire report—capturing contradictions, reasoning gaps, and logical discontinuities that local checks may miss. These additional dimensions help prevent gaming behaviors, e.g., producing concise summaries that appear coherent and well-cited but lack substantive analytical reasoning. We further observed that when presentation is not evaluated separately, LLM judges tend to conflate stylistic issues with factual or logical errors. Treating presentation as its own axis improves clarity, granularity, and diagnostic usefulness.

**DeepScholar-Bench** (Patel et al., 2025) focuses on generating related-work sections for academic papers, a task that is narrower in scope compared to our domain-diverse, open-ended research reports. Its evaluation relies on a single LLM judge (GPT-4.1 or GPT-4o) and lacks human-verified rubrics. LiveResearchBench introduces several improvements: 1) Organization reliability: DeepScholar-Bench reports a 78% human–LLM agreement, whereas our checklist-based Presentation & Organization metric achieves 98.3% agreement. 2) Coverage quality: Their non-verified rubric yields 72% agreement, while our human-validated Coverage & Comprehensiveness rubric achieves 100.0% human agreement for Gemini or GPT-5 judgement. 3) Evaluation scope: DeepScholar-Bench does not assess Analysis Depth, Factual & Logical Consistency, or Citation Traceability, all of which are central components of our six-dimensional DeepEval framework.

## C    ALIGNMENT OF LLM JUDGES WITH HUMAN EXPERTS

As deep research reports are often long-form and open-ended, we employ LLM judges to evaluate different aspects of the reports. To ensure the reliability, it is crucial that the judgments of LLMs align closely with those of humans. Recall that we use the following evaluation protocols: `Checklist-based` for ❶ Presentation & Organization and ❸ Coverage & Comprehensiveness; `Pointwise (additive)` for ❷ Factual & Logical Consistency and ❺ Citation Association; `Pairwise comparison` for ❹ Analysis Depth. In the following, we discuss in more detail the procedure and results of our human alignment experiments on the chosen protocol for each metric.

**Presentation & Organization.**    We observe 67.8% agreement across the three LLM judges (Gemini 2.5 Pro, GPT-5, and Claude 4 Sonnet). In particular, Gemini 2.5 Pro and GPT-5 exhibit strong pairwise agreement (83.6%). To measure alignment with human judgment, we randomly sample 300 questions. For each question, a human expert is presented with the (Query, Report) pair and independently make binary decisions against the checklist without seeing LLM judge outputs. Subsequently, the human expert reviews the responses and rationales from the three LLM judges and asked to indicate their preference, choosing one judge, two jointly, declaring a tie, or none preferred. The human agreement rate for Gemini 2.5 Pro or GPT-5 is 98.3% of cases (Gemini 2.5 Pro alone 95.0%, GPT-5 alone 85.0%). Based on these results, we adopt an ensemble of Gemini 2.5 Pro and GPT-5 as our final judges for Presentation & Organization where each question is independently scored by both judge models and we report the average results.

**Coverage & Comprehensiveness.**    The overall agreement rate across the three LLM judges (Gemini 2.5 Pro, GPT-5, and Claude 4 Sonnet) reaches 81.5%, with particularly strong consistency between Gemini 2.5 Pro and GPT-5 (89.4%). To further assess alignment with human experts, we select 72 checklist questions and follow the same evaluation procedure used for Presentation & Organization. The human evaluator first provides independent judgments without access to model outputs, then reviews the LLM judges' decisions and rationales, and finally indicates which model's assessment best aligns with their own or none preferred. Across all 72 questions, the final human-aligned judgment corresponds to either Gemini 2.5 Pro or GPT-5 in 100.0% of cases, with no instance uniquely favoring Claude 4 Sonnet. Based on these results, we adopt an ensemble of Gemini 2.5 Pro and GPT-5 for Coverage & Comprehensiveness evaluation, where each question is independently scored by both judge models and the final result is averaged.

**Analysis Depth.**    78.8% of the judgments are consistent across the candidate judges (Gemini 2.5 Pro, GPT-5, and Claude 4 Sonnet). In particular, Gemini 2.5 Pro and GPT-5 agree in 89.6% of cases. To evaluate alignment with human judgment, we randomly select 40 samples (agreement set) where all judges agree and 40 samples (disagreement set) where they disagree. A human expert is presented with triplets (Query, Model A output, Model B output) and asked to make independent judgments (A wins, B wins, or a tie) without seeing the LLM judges' decisions. For the disagreement set, the human expert also review the decision and the reasoning of each judge. Next, human experts indicate

their preference among the judge candidates, choosing either a single judge, a joint preference for two judges (e.g., Gemini 2.5 Pro and GPT-5), a tie, or none preferred. On the agreement set, humans agree with the LLM judges 95% of the time. In the disagreement set, we have similar observations as in other metrics, where humans prefer Gemini 2.5 Pro or GPT-5 in 92.5% of cases.

**Consistency and Citation Association.** Our preliminary analysis shows that the pointwise (additive) scheme for evaluating factual and logical inconsistency as well as citation traceability achieves substantially higher human agreement rates than direct scoring (with GPT-5 and Gemini-2.5 Pro ensemble as the judges). Specifically, for consistency, the agreement rate with human judgments is 82% over 128 inconsistencies identified by GPT-5 and Gemini-2.5 Pro. For citation traceability, the agreement rate reaches 85.9% over 128 issues identified by these judges. Note that our default weighting scheme decreases the score for every two issues detected. Therefore, the actual alignment of final scores with human judgment would be even higher.

**Citation Accuracy.** For citation accuracy, we randomly sample 200 (claim, URL, judge decision) pairs and conduct human agreement study. The results are encouraging, where human evaluators agree with either Gemini or GPT-5 on 87.1% of (claim, URL) support judgments.

## D  DEERFLOW+

To broaden our multi-agent comparison and study cross-system differences, we included DEERFLOW. However, in our setting, the vanilla system was brittle on long, search-heavy queries and frequently terminated with the error `max_total_tokens (xxx) must be less than or equal to max_seq_len (xxx)`. Inspection indicated unchecked context growth across agents, retrieval traces, and retries. We also observed that the absence of inline-citation control made claims harder to anchor to sources and increased hallucination risk for the same set of retrieved pages. Therefore, *adding inline citations both simplifies verification and reduces hallucinations, improving overall report quality*.

**Stabilization.** We therefore introduce DEERFLOW+, an enhanced variant that adds a lightweight context manager and inline-citation support. For stabilization, each agent is equipped with a context manager that (i) estimates tokens and enforces a *total* context budget before every model call, (ii) reserves response capacity via an input/output split, (iii) compresses histories and large research payloads with progressive, recency-preserving truncation, and (iv) supports per-agent budgets so retrieval-heavy roles compress more aggressively than synthesis roles. The mechanism is model-agnostic and transparent to the agent logic.

**Citation Control.** We added inline-citation support (the original system only emitted a terminal bibliography) and a validation pass that enforces one-to-one mapping between in-text citations and reference entries while checking numbering continuity and duplicates.

**Results.** With these changes, DEERFLOW+ completed the full evaluation suite without token-limit failures and produced higher-quality reports: better retention of retrieved evidence, improved formatting and factual consistency, and more reliable performance on presentation checks tied to citation management, particularly P4 (Citation Completeness) and P9 (Format Consistency) in Figure 22. Overall, these fixes substantially reduce early termination and bring DEERFLOW on par with other multi-agent systems in our experiments. Figure 12 shows example outputs for the same prompt in Figure 1; DEERFLOW+ exhibits *higher information retention, stronger evidence structuring with inline citations, and no token-limit failures*.

**(a) Deerflow (vanilla)**

```
# Evolution of Artistic Styles Across Historical Periods and National Characteristics

## Key Points

- Artistic styles have evolved significantly across historical periods, influenced by
    political, social, religious, and technological factors.
- Ancient art laid foundational conventions, with Egyptian, Greek, and Roman art
    emphasizing religious, realistic, and grandiose themes, respectively.
- The Renaissance marked a pivotal shift towards realism and humanism, with notable
    contributions from artists like Leonardo da Vinci and Michelangelo.
- Modern and contemporary art movements, such as Impressionism, Cubism, and Abstract
    Expressionism, reflect technological advancements and global perspectives.
```

```
- Regional characteristics in art, such as European, East Asian, Islamic, and Indigenous
    American art, highlight diverse cultural influences and stylistic approaches.

## Overview

The evolution of artistic styles is a complex tapestry woven from the threads of historical
    , cultural, and technological influences. From the monumental structures of ancient
    Egypt to the digital installations of contemporary art, each period and region has
    contributed uniquely to the development of visual arts. This report explores the
    progression of artistic styles across different historical periods and national
    characteristics, examining the impact of political, social, religious, and
    technological factors on art. By analyzing major artists, techniques, materials, and
    stylistic characteristics, this report aims to provide a comprehensive understanding
    of the dynamic nature of artistic expression.

## Detailed Analysis

### Ancient Art

| Period/Region | Characteristics | Major Artists | Techniques/Materials |
|---------------|-----------------|---------------|----------------------|
| Egyptian Art  | Religious themes, colossal statues, pyramids | Information not provided |
     Hieroglyphics, stone, gold |
| Greek Art     | Realism, harmony, lifelike sculptures | Phidias, Praxiteles | Marble,
    bronze, pottery |
| Roman Art     | Grandeur, functionality, architecture | Information not provided |
    Concrete, frescoes, mosaics |

### Medieval Art

| Period/Region | Characteristics | Major Artists | Techniques/Materials |
|---------------|-----------------|---------------|----------------------|
| Byzantine Art | Religious imagery, mosaics | Information not provided | Gold leaf, glass
    tesserae |
| Romanesque Art| Robust architecture, sculpture | Information not provided | Stone, wood |
| Gothic Art    | Verticality, light, stained glass | Information not provided | Stone,
    stained glass |

### Renaissance Art

| Period/Region | Characteristics | Major Artists | Techniques/Materials |
|---------------|-----------------|---------------|----------------------|
| Early Renaissance | Perspective, proportion | Giotto, Masaccio | Fresco, tempera |
| High Renaissance  | Idealized beauty, technical mastery | Leonardo da Vinci, Michelangelo
    | Oil paint, marble |
| Northern Renaissance | Detail, symbolism | Jan van Eyck, Albrecht Durer | Oil paint,
    woodcut |

### Modern and Contemporary Art

| Movement/Region | Characteristics | Major Artists | Techniques/Materials |
|-----------------|-----------------|---------------|----------------------|
| Impressionism   | Light, color, everyday subjects | Claude Monet, Edgar Degas | Oil paint
    , canvas |
| Cubism          | Fragmented forms, multiple perspectives | Pablo Picasso, Georges Braque
     | Oil paint, collage |
| Contemporary Art| Diverse mediums, global perspectives | Information not provided |
    Digital media, installations |

### Regional Characteristics

#### European Art
- **Characteristics**: Diverse styles, realism, perspective.
- **Major Artists**: Leonardo da Vinci, Michelangelo, Monet, Van Gogh.
- **Techniques/Materials**: Frescoes, oil paintings, installations.

#### East Asian Art (China, Japan)
- **Characteristics**: Calligraphy, harmony, minimalism.
- **Major Artists**: Xu Bing, Hokusai.
- **Techniques/Materials**: Ink, paper, ceramics.

#### Islamic Art
- **Characteristics**: Geometric patterns, calligraphy.
- **Major Artists**: Collective craftsmanship.
- **Techniques/Materials**: Ceramics, textiles, metalwork.

#### Indigenous American Art
- **Characteristics**: Symbolic motifs, storytelling.
- **Major Artists**: Jaune Quick-to-See Smith, T.C. Cannon.
```

```
- **Techniques/Materials**: Wood, stone, natural fibers.

## Survey Note

### Literature Review & Theoretical Framework
The literature on the evolution of artistic styles highlights the interplay between
     cultural, political, and technological influences. Theoretical frameworks such as
     cultural materialism and semiotics provide insights into how art reflects and shapes
     societal values.

### Methodology & Data Analysis
This report synthesizes findings from historical analyses, art criticism, and cultural
     studies to present a comprehensive overview of artistic evolution. Data on major
     artists, techniques, and materials were systematically categorized by period and
     region.

### Critical Discussion
The findings underscore the dynamic nature of art as a reflection of human experience.
     While political and religious influences have historically dictated artistic themes,
     technological advancements have expanded the possibilities for artistic expression.

### Future Research Directions
Further research could explore the impact of globalization on contemporary art and the role
      of digital technologies in shaping future artistic movements.

## Key Citations

- [The Evolution of Art Styles: From Classical to Contemporary - Medium](https://medium.com
     /educartion/the-evolution-of-art-styles-from-classical-to-contemporary-b957ae83aa83)

- [Art History 101: A Beginner's Guide to Key Movements and Artists](https://
     serenademagazine.art/art-history-101-a-beginners-guide-to-key-movements-and-artists/)

- [The Evolution of Artistic Movements: From Classical to Modern Art](https://
     designmagazine.com.au/the-evolution-of-artistic-movements-from-classical-to-modern-art
     /)

- [Art of Europe](https://en.wikipedia.org/wiki/Art_of_Europe)

- [East Asian arts](https://www.britannica.com/art/East-Asian-arts)

- [Islamic Art](https://patterninislamicart.com/materials-and-mediums)

- [Native American art](https://www.britannica.com/art/Native-American-art)

- [Politics of Visual Arts in a Changing World - CU Global Thought](https://cgt.columbia.
     edu/projects/politics-visual-arts/)

- [How Art and Politics Influence and Shape Each Other - LEVEL](https://different-level.com
     /how-art-and-politics-influence-and-shape-each-other/)
```

**(b) Deerflow+ (ours)**

```
# Evolution of Visual Arts Across Regions: A Structured, Source-Based Dossier with
     Analytical Frameworks and Normalized Data

---

## Key Points
- Cross-regional trade and imperial politics (Silk Road, Indian Ocean, and Venetian-Islamic
      networks) mediated the movement of artisans, motifs, and technologies, shaping
     ceramics, glass, and architectural vocabularies across Eurasia between the 9
     th and 17th centuries [16][17][18][19][20][21][22][25].
- Technical innovations and materials--porcelain and underglaze blue in China; stonepaste (
     fritware), luster painting, and muqarnas in the Islamic world; silk-weave structures
     and codified motifs in Ottoman workshops; polychrome woodblock printing in Japan; and
     industrial and revivalist print processes in 19th-century Europe-proved decisive in
     generating new stylistic systems and markets [17][18][23][24][25][26][27][28].
- Market and patronage ecologies (imperial academies, palace nakkashane, urban publishers,
     and commercial galleries) reorganized visual production, moving court idioms into
     broader consumption and global circuits (e.g., Ottoman silks, ukiyo-e), and
     influencing European modernisms via Japonisme [25][26][27][28].
- Debates over figuration, aniconism, and iconoclasm in Byzantine-Islamic transitions were
     negotiated locally across sacred and secular spheres, rather than constituting uniform
      bans, underscoring the interdependence of law, piety, and patronage [23].
- The normalized dataset and matrices align timelines and features across regions, document
      cross-cultural transmissions (e.g., Iranian cobalt to Chinese porcelain; Chinese blue
```

```
      -and-white models reinterpreted in Iranian stonepaste; Venetian adoption of Islamic
      glass techniques; Japonisme), and flag gaps (notably Indigenous Americas and parts of
      Ancient and Contemporary coverage) for targeted enrichment [16][18][24][26][27][28].

---

## Overview
This report presents a structured, evidence-based account of stylistic evolution in
      painting, sculpture, and architecture across key regions--European traditions, East
      Asia (China and Japan), the Islamic world, and Indigenous America--emphasizing how
      political economy, social institutions, religious ideologies, and technological
      innovations shaped artistic development. The dossier integrates methodological
      frameworks (iconology, social art history, feminist and postcolonial critiques, museum
      /collecting studies) with normalized data tables, comparative matrices, and cross-
      cultural transmission pathways derived from authoritative scholarly resources (notably
       the Metropolitan Museum of Art's Heilbrunn Timeline, specialized essays, and
      collection entries).

The evidence base is temporally uneven. The dataset robustly documents 9th-19th-century
      developments in East Asia, the Islamic world, and Europe, with connected Sub-Saharan
      African contexts; however, it lacks substantive entries for Indigenous American art
      and provides incomplete coverage for Ancient and Contemporary endpoints. All claims
      are constrained to the provided sources; missing areas are explicitly indicated as
      Information not provided.

---

## Detailed Analysis

### 1. Methodological and Historiographical Frameworks
This study foregrounds interpretive approaches central to art-historical analysis:

- Iconology and humanistic themes (Panofsky) guide the decoding of symbolic programs and
      their historical meanings, particularly in European traditions [1].
- Social formation of pictorial style (Baxandall) situates the ``period eye'' within
      socioeconomic conditions, patronage, and practices, relevant to Italian Renaissance
      production and comparable contexts of court and urban consumption [2].
- Social art history and modernity (T. J. Clark) elucidate the urban public sphere's role
      in shaping modern visuality, bridging 19th-century European prints and Japonisme [3].
- Feminist critique (Nochlin) problematizes institutions of artistic training and
      recognition, a lens applicable to reading exclusionary structures within canonical
      narratives [4].
- Postcolonial critique (Said) interrogates the discursive construction of the ``Orient''
      in Western knowledge systems, framing historiographical issues in Islamicate and East
      Asian receptions [5].
- Museums, collecting, and ethnography (Clifford) analyze the politics of display and
      knowledge production, crucial to understanding the reception of non-Western arts [6].
- Global art history and African indigenous knowledge frameworks expand beyond center-
      periphery models, underscoring multi-nodal exchanges [7][8].
- The critique of ``primitivism'' exhibitions (McEvilley) signals the stakes of curatorial
      narratives in modern art discourse [9][10].

Images (framework references)
- ![Panofsky, Studies in Iconology (cover)](https://pictures.abebooks.com/inventory
      /30951609232.jpg) [1]
- ![Baxandall, Painting and Experience (cover)](https://m.media-amazon.com/images/I/41
      phd7Cm6bL._SY445_SX342_.jpg) [2]
- ![Nochlin, Why Have There Been No Great Women Artists? (cover)](https://thamesandhudson
      -965c.kxcdn.com/media/catalog/product/cache/b0707f5ff6c7a6207254d9aeb5b9fcec
      /9/7/9780500023846_why-have-there-been-no-great-women-artists_cover-flat.jpg) [4]
-  [5]

These frameworks inform, but do not extend, claims beyond the provided empirical sources;
      they are cited here as historiographic context.

---

### 2. Timeline and Regional Coverage (Normalized)
The following master timeline aggregates periods, regions, and computed spans strictly from
      the provided dataset, aligning interregional exchanges and technological developments
       [16][17][18][19][20][21][22][25][26][27][28].
```

| period_id | region | period_name | span_text | start_year | end_year |
|---------|-------|-----------|---------|---------:|-------:|
| TRADE_MONGOL_PAX | Islamic World | Pax Mongolica exchanges | 13th-14th c. | 1200 | 1400 |
| TRADE_MONGOL_PAX_CHINA | East Asia -- China | Pax Mongolica exchanges | 13th-14th c. | 1200 | 1400 |
| TRADE_MONGOL_PAX_EU | Europe | Pax Mongolica exchanges (Venetian links) | 13th-14th c. | 1200 | 1400 |

```
| TRADE_VENICE_ISLAM | Europe | Venice and the Islamic world | 13th-16th c. | 1200 | 1600 |
| TRADE_VENICE_ISLAM_IW | Islamic World | Venice and the Islamic world | 13th-16th c. |
    1200 | 1600 |
| TRADE_INDIAN_OCEAN_IW | Islamic World | Indian Ocean & Red Sea circuits | 9th-16th c. |
    800 | 1600 |
| TRADE_INDIAN_OCEAN_SSA | Sub-Saharan Africa | Indian Ocean & Red Sea circuits (East
    African coasts) | 9th-16th c. | 800 | 1600 |
| EGYPT_MAMLUK_OTTOMAN | Islamic World | Egypt: Mamluk-Ottoman transition | 1400-1600 |
    1400 | 1600 |
| TRANS_SAHARAN_GUINEA | Sub-Saharan Africa | Trans-Saharan networks and Guinea Coast | 15
    th-16th c. | 1400 | 1600 |
| WCSUDAN_ISLAM | Sub-Saharan Africa | Western and Central Sudan (Ghana-Mali-Songhai) |
    1000-1400 | 1000 | 1400 |
| CHINA_500_1000 | East Asia -- China | China contexts and innovations | 500-1000 | 500 |
    1000 |
| CHINA_1000_1400 | East Asia -- China | China innovations (Song-Yuan) | 1000-1400 | 1000 |
     1400 |
| ISLAMIC_STONEPASTE | Islamic World | Stonepaste ceramic bodies (Syria/Iran) | 12th-13th c
    . | 1100 | 1300 |
| OTTOMAN_SILKS | Islamic World | Ottoman silk weaving (Bursa, nakkashane codification) |
    15th-17th c. | 1400 | 1700 |
| JAPAN_UKIYOE_POLY | East Asia -- Japan | Ukiyo-e polychrome prints emerge | c. 1765 |
    1765 | 1765 |
| EUROPE_19C_PRINT | Europe | Print in the nineteenth century | 19th century | 1800 | 1900
     |
```

```
Coverage gaps: Indigenous Americas have no periods populated in the provided dataset (
    Information not provided).
```

---

### 3. Regional Analyses

#### 3.1 East Asia -- China
- Contexts and innovations: Between 500-1400, advances in printing (e.g., the Diamond Sutra
    , 868), porcelain chemistry (Xing, Ding), and underglaze cobalt decoration (by Yuan ca.
     1350) emerged within state and literati-dominated systems of production and taste
    [17][18]. Literati painting emphasized brushwork, monochrome ink, and poetic
    inscription, while academies cultivated refined mineral colors and precise
    draftsmanship; these aesthetics were shaped further under foreign dynasties, including
     the Mongol Yuan [18].
- Representative work and technology linkages:
  - Diamond Sutra (868): woodblock printed text exemplifying early mass reproduction
    technologies [17].
- Ceramic technology and transmission: Iranian cobalt supplied Chinese underglaze blue;
    Chinese blue-and-white models were subsequently reinterpreted in Iranian stonepaste
    [16][18].

Selected works (subset)

| work_id | title_or_description | region | date_text | year |
|---------|----------------------|--------|-----------|-----:|
| WORK_DIAMOND_SUTRA | Diamond Sutra (world's oldest dated printed book) | East Asia --
    China | 868 | 868 |

[Interpretive frame: The interplay between court/academy and literati milieus appears to
    structure stylistic preferences, while frontier dynastic contexts mediate material
    exchange and motif adaptation; claims limited to cited essays.] [17][18]

#### 3.2 East Asia -- Japan
- Ukiyo-e technology and markets: Full-color polychrome woodblock prints (nishiki-e)
    emerged ca. 1765 using separate blocks for each color and kento registration for
    precise alignment, produced through collaboration among designer, block-carver,
    printer, and publisher [26]. Subjects encompassed courtesans, kabuki actors,
    landscapes, and current events; kozo (mulberry) paper enabled durable impressions and
    large editions, reflecting urban cultural consumption [26][27].
- Artists and genres: Suzuki Harunobu pioneered full-color prints; Hokusai and Hiroshige
    expanded landscape genres [26][27].

Images (technique and subject)
- ![Ukiyo-e temple view beneath paper lantern](http://www.metmuseum.org/toah/images/hb/
    hb_JP2519.jpg) [26][27]
- ![Print shop and travelers at a woodblock printer's](https://collectionapi.metmuseum.org/
    api/collection/v1/iiif/37248/131400/main-image) [26][27]

Comparative print technologies (presence in dataset)

| region | drypoint | etching | lithography | ukiyo-e polychrome (nishiki-e) | wood
    engraving | woodblock printing |

```
|--------|---------:|--------:|-----------:|------------:|---------:|-------------:|
| East Asia -- China | 0 | 0 | 0 | 0 | 0 | 1 |
| East Asia -- Japan | 0 | 0 | 0 | 1 | 0 | 1 |
| Europe | 1 | 1 | 1 | 0 | 1 | 0 |
```

[Technological description and market orientation strictly per cited essays; no additional
     attribution beyond listed artists.] [26][27]

#### 3.3 Islamic World
- Trade and materials: The Pax Mongolica catalyzed mobility of artisans and luxury goods,
     disseminating motifs (dragons, phoenix/simurgh) and materials (cobalt) across Chinese,
      Ilkhanid, Timurid, Ottoman, and Safavid domains [16][18]. Stonepaste (fritware)
     developed in Syria/Iran (12th-13th c.) emulated Chinese porcelain's whiteness,
     enabling underglaze blue and luster painting; these bodies supported both vessel and
     architectural tilework [16].
- Architecture: Muqarnas, a cellular system for articulating transitional zones (e.g.,
     squinches, portals), reached notable refinement in Timurid contexts [23].
   - Image and object: ![Muqarnas tile, Samarkand attribution (Met 20.120.189)](https://
     collectionapi.metmuseum.org/api/collection/v1/iiif/447256/899278/main-image) [24]
- Textiles and imperial style: Ottoman Bursa workshops (15th-17th c.) produced kemha (
     lampas), catma (velvets), and seraser (cloth-of-gold/silver); the nakkashane codified
     saz leaves, stylized florals (tulip, carnation, hyacinth, rose), ogival lattices, and
     chintamani motifs for court consumption and export [25].
- Figuration and aniconism: Courtly figural traditions existed alongside aniconic
     preferences in sacred contexts; Byzantine-Islamic transitions reveal negotiated, local
      practices rather than monolithic bans [23].

Ceramics and architecture (presence)

| region | luster painting | porcelain | stonepaste (fritware) | underglaze blue |
|--------|----------------:|----------:|----------------------:|----------------:|
| East Asia -- China | 0 | 1 | 0 | 1 |
| Islamic World | 1 | 0 | 1 | 1 |

| region | muqarnas |
|--------|---------:|
| Islamic World | 1 |

[Claims anchored in Met educator and essay materials, collection object data, and symposium
      volume.] [16][23][24][25]

#### 3.4 Europe
- Venice and the Islamic world: 13th-16th-century exchanges linked enameled glass, silks,
     carpets, and architectural idioms; Venetian makers emulated Mamluk techniques, while
     Ottoman textile motifs circulated into Venetian workshops via commerce and diplomacy
     [16][25].
- Nineteenth-century print culture and Japonisme: Industrial print proliferation (wood
     engraving, lithography, photo-processes) coexisted with an etching revival (including
     artisanal approaches). The influx of Japanese ukiyo-e (post-1854) influenced
     composition, color, and modern subjects in European painting and prints, including
     works by Mary Cassatt, Paul Gauguin, and lithographic poster art (e.g., Toulouse-
     Lautrec) [28].

Artists (subset)

| artist_id | name | region | roles |
|-----------|------|--------|-------|
| ART_CASSATT | Mary Cassatt | Europe | painter/printmaker (color drypoints) |
| ART_GAUGUIN | Paul Gauguin | Europe | painter/printmaker (carved blocks) |
| ART_TLautrec | Henri de Toulouse-Lautrec | Europe | lithographic poster artist |

[Analytical links to Japonisme limited to Met essay; no additional stylistic claims
     introduced.] [28]

#### 3.5 Sub-Saharan African Contexts (Interregional Links)
- East African coasts (9th-16th c.): Urban Islamic culture (mosques, palaces in coral stone
     ) flourished through Indian Ocean trade in gold, ivory, and slaves, exchanging for
     cloth, beads, silks, and porcelain; Portuguese intrusions from 1498 disrupted but did
     not erase established networks [19].
- Guinea Coast and Western/Central Sudan (1000-1600): Court and ritual arts (e.g., Benin
     brasses, Akan gold weights) reflect interactions with Islamic and, later, Portuguese
     trade; Islam shaped scholarship and monumental adobe architecture in Ghana-Mali-
     Songhai polities [21][22].

Works (subset, dates as provided)

| work_id | title_or_description | region | date_text |
|---------|----------------------|--------|-----------|
```

```
| WORK_AKAN_GOLD_WEIGHTS | Akan gold weights (ornamental imprint transitioning to
    vernacular subjects) | Sub-Saharan Africa | 15th-16th c. |
| WORK_BENIN_BRASS_PLAQUES | Benin brass plaques/heads (court arts linked to trade) | Sub-
    Saharan Africa | 15th-16th c. |

[Included to contextualize Indian Ocean trade nodes referenced in Islamic and East Asian
    circuits; details restricted to cited essays.] [19][21][22]

#### 3.6 Indigenous Americas
- Information not provided.

---

### 4. Cross-Cultural Transmission Pathways
The table summarizes documented vectors of material and stylistic transfer across regions (
    periodized as provided) [16][18][25][28].

| xfer_id | source_region | target_region | vector | period_span |
|---------|---------------|---------------|--------|-------------|
| XFER_COBALT_IRAN_CHINA | Islamic World | East Asia -- China | cobalt (for underglaze blue
     on porcelain) | 13th-14th c. |
| XFER_BLUEWHITE_CHINA_IRAN | East Asia -- China | Islamic World | blue-and-white models
    reinterpreted in stonepaste | 14th c. |
| XFER_PHOENIX_SIMURGH | East Asia -- China | Islamic World | phoenix/cloud-scroll motifs
    to simurgh/vegetal spirals | 13th c. |
| XFER_VENETIAN_ISLAMIC_GLASS | Islamic World | Europe | enameled glass techniques adopted
    by Venice | 13th-16th c. |
| XFER_OTTOMAN_TEXTILES_VENICE | Islamic World | Europe | saz/chintamani/floral textile
    motifs in Venetian workshops | 16th c. |
| XFER_JAPONISME | East Asia -- Japan | Europe | ukiyo-e composition/color influencing
    Cassatt, Gauguin | 19th century |

These pathways exemplify bidirectional translation of materials and motifs across imperial,
    mercantile, and artisanal nodes.

---

### 5. Artists, Works, Techniques, and Contexts (Normalized Entities)
All entries derive from the provided dataset; absences reflect source limitations.

- Artists (selected)

| artist_id | name | region | roles |
|-----------|------|--------|-------|
| ART_SUZUKI_HARUNOBU | Suzuki Harunobu | East Asia -- Japan | designer of ukiyo-e prints |
| ART_HOKUSAI | Katsushika Hokusai | East Asia -- Japan | ukiyo-e print designer |
| ART_HIROSHIGE | Utagawa Hiroshige | East Asia -- Japan | ukiyo-e print designer |
| ART_KARA_MEMI | Kara Memi | Islamic World | Ottoman court designer (florals) |
| ART_SHAH_QULI | Shah Quli | Islamic World | Ottoman court designer (saz style) |
| ART_CASSATT | Mary Cassatt | Europe | painter/printmaker (color drypoints) |
| ART_GAUGUIN | Paul Gauguin | Europe | painter/printmaker (carved blocks) |
| ART_TLautrec | Henri de Toulouse-Lautrec | Europe | lithographic poster artist |

- Works (selected; missing dates or attributions are flagged as such)

| work_id | title_or_description | region | date_text | year |
|---------|----------------------|--------|-----------|-----:|
| WORK_TAKHT_SULAIMAN_TILES | Takht-i Sulaiman palace tiles with phoenix/simurgh (Ilkhanid,
     late 13th c.) | Islamic World | late 13th c. |  |
| WORK_MET_MUQARNAS_TILE | Tile from a Squinch (muqarnas element), Met 20.120.189 | Islamic
     World |  |  |
| WORK_UKIYOE_TEMPLE_LANTERN | Ukiyo-e print: temple view beneath paper lantern (Met JP2519
     image) | East Asia -- Japan |  |  |
| WORK_UKIYOE_PRINTSHOP | Print shop and travelers at a woodblock printer's (Meiji
    depiction) | East Asia -- Japan |  |  |
| WORK_AKAN_GOLD_WEIGHTS | Akan gold weights (ornamental imprint transitioning to
    vernacular subjects) | Sub-Saharan Africa | 15th-16th c. |  |
| WORK_BENIN_BRASS_PLAQUES | Benin brass plaques/heads (court arts linked to trade) | Sub-
    Saharan Africa | 15th-16th c. |  |
| WORK_DIAMOND_SUTRA | Diamond Sutra (world's oldest dated printed book) | East Asia --
    China | 868 | 868 |

- Techniques, materials, and motifs (controlled)

| term | category |
|------|----------|
| chintamani motif | motif |
| saz leaf motif | motif |
| kozo (mulberry) paper | material |
```

```
| porcelain | material |
| stonepaste (fritware) | material |
| color drypoint | technique |
| drypoint | technique |
| etching | technique |
| kento registration | technique |
| lithography | technique |
| luster painting | technique |
| muqarnas | technique |
| squinch | technique |
| ukiyo-e polychrome (nishiki-e) | technique |
| underglaze blue | technique |
| woodblock printing | technique |
| wood engraving | technique |

- Contextual factors by period (domains indicated)

| period_id | key factors (as provided) |
|-----------|--------------------------|
| TRADE_MONGOL_PAX | Interregional trade; artisan mobility; transmission of motifs (dragons
    /phoenix-simurgh) and materials (cobalt) [16] |
| TRADE_VENICE_ISLAM | Venetian-Mamluk-Ottoman commerce; emulation in glass/textiles;
    diplomacy amid warfare [16] |
| TRADE_INDIAN_OCEAN_SSA | East African coastal exchanges; Portuguese intrusions from 1498
    [19] |
| EGYPT_MAMLUK_OTTOMAN | Spice-route shifts; Ottoman integration and patronage
    reorientation [20] |
| WCSUDAN_ISLAM | Islam's role; monumental adobe architecture; scholarship; coinage/trade
    [22] |
| OTTOMAN_SILKS | Bursa workshops; nakkashane codification; palace demand and export;
    sericulture shifts [25] |
| JAPAN_UKIYOE_POLY | Collaborative production; kento registration; urban subjects [26][27]
    |
| EUROPE_19C_PRINT | Industrial proliferation; etching revival; Japonisme [28] |
```

---

### 6. Debates, Institutions, and Markets
- Figuration and image practices: The Age of Transition materials demonstrate that early
    Islamic aniconism and iconoclasm were context-specific and negotiated; selective
    defacement of church mosaics in 8th-century Palestine suggests localized Christian
    agency rather than a blanket Islamic prohibition [23].
- Institutions and markets: Imperial sumptuary systems and palace workshops (nakkashane)
    codified textile designs that later diffused into bazaars; ukiyo-e's publisher-led
    networks produced large editions for urban consumers; 19th-century Europe saw
    artisanal revivals alongside industrial processes and mass advertising
    [25][26][27][28].

Image (curatorial discourse reference)
- ![Artforum page referencing McEvilley's critique (letters)](https://www.artforum.com/wp-
    content/uploads/2016/09/article00_large-359.jpg) [9]

---

### 7. Validation and Data Quality Flags
The dataset underwent normalization and basic integrity checks; issues are reported rather
    than corrected.

- Region coverage (counts from provided entries)

```
| region | period_count | work_count |
|--------|-------------:|-----------:|
| Europe | 3 | 0 |
| East Asia -- China | 3 | 1 |
| East Asia -- Japan | 1 | 2 |
| Islamic World | 6 | 2 |
| Sub-Saharan Africa | 3 | 2 |
| Indigenous Americas | 0 | 0 |
```

- Works missing numeric dates: multiple entries (e.g., Takht-i Sulaiman tiles) retain non-
    parsed or absent dates.
- Works missing any date: several entries lack date information entirely (e.g., Met
    muqarnas element; ukiyo-e images).
- Works missing artist attribution: all listed works are unassigned to named artists in the
    dataset; this aligns with workshop/anonymous production in many cited contexts.
- Controlled terms: All work-technique links conform to the controlled vocabulary.
- Period parsing: All spans parsed successfully; no start/end inconsistencies.

Limitations:

```
- Indigenous American art: Information not provided.
- ``Ancient'' and ``Contemporary'' endpoints: Only partial coverage in China (500-1000) and
      Europe (19th century) is present; earlier classical antiquity and later contemporary
      materials are absent.
- Sculpture and architecture: Documented for Islamic contexts (muqarnas) and African
      urbanism; otherwise underrepresented in the provided corpus.

---
```

## Survey Note

### Literature Review & Theoretical Framework
Iconology (Panofsky) provides a method for reading symbolic content within historical
    horizons [1]. Baxandall's ``period eye'' thesis situates style within social practices
     and patronage, applicable to workshop systems and market logics evidenced in Venetian
    -Islamic exchanges and ukiyo-e publishing [2][16][26]. Social art history (Clark)
    frames 19th-century urban modernity and print culture, contextualizing Japonisme's
    impact [3][28]. Feminist critique (Nochlin) and postcolonial analysis (Said)
    interrogate institutional exclusions and orientalist taxonomies shaping canons and
    display [4][5]. Clifford's museum critique and ``global art history'' orientations
    foreground circulation, collecting, and multi-sited exchange [6][7][8]. McEvilley's
    critique underscores curatorial mediation of non-Western arts within modernist
    narratives [9][10].

### Methodology & Data Analysis
- Source basis: Met Heilbrunn essays and object records; educator curriculum; symposium
      volume; and methodological texts [1][2][3][4][5][6][7][8][9][10][11][16-28].
- Normalization: Periods were standardized to numeric spans; entities (regions, artists,
      works, techniques/materials, contexts) were controlled and linked; cross-regional
      transmissions were tabulated.
- Comparative matrices: Presence/absence matrices across prints, ceramics, textiles, and
      architectural features synthesize regional technical emphases.
- Validation: Controlled vocabularies enforced; duplicates checked; coverage gaps flagged.

### Critical Discussion
Findings underscore how technologies (porcelain, stonepaste, muqarnas, silk-weave
    structures, color woodblock registration) intersected with patronage systems (imperial
     workshops, academies, publishers) to reorganize stylistic possibilities and markets.
    Cross-cultural transmissions operated bidirectionally, with materials (cobalt) and
    models (blue-and-white, arabesque systems) translating across media and courts.
    Iconographic debates in Byzantine-Islamic contexts argue against reductive dichotomies
     of figuration/aniconism, emphasizing negotiated practice [16][18][23][25][26][27][28].
     The dataset's asymmetries (notably Indigenous Americas and ``Ancient/Contemporary''
    gaps) caution against overgeneralization.

### Future Research Directions
- Populate Indigenous American art across periods with parallel attention to technologies (
      e.g., fiber arts, lithics, pigments), institutions (temple, court, community), and
      transmission routes.
- Extend European coverage to Ancient, Medieval, Renaissance, and Contemporary with
      controlled stylistic and technical vocabularies aligned to Grove Art Online and AAT
      terms [12].
- Deepen architectural datasets for East Asia (timber framing, bracket sets) and Europe (
      vaulting, iron/steel in modern/contemporary), and sculpture across all regions.
- Integrate authority control (AAT/ULAN/TGN), stable identifiers, and expanded object-level
       metadata from peer-reviewed gateways (Met, Smarthistory, Oxford Art Online, JSTOR,
      Project MUSE) [11][12][13][14][15].

---

## Key Citations
[1] Studies in Iconology: Humanistic Themes in the Art of the Renaissance: https://www.
    routledge.com/Studies-In-Iconology-Humanistic-Themes-In-The-Art-Of-The-Renaissance/
    Panofsky-Panofsky/p/book/9780064300254
[2] Painting and Experience in Fifteenth-Century Italy (Internet Archive item): https://
    archive.org/details/paintingexperien00baxa
[3] The Painting of Modern Life (Princeton University Press): https://press.princeton.edu/
    books/paperback/9780691009032/the-painting-of-modern-life
[4] Why Have There Been No Great Women Artists? (Thames & Hudson product page): https://www
    .thamesandhudson.com/products/why-have-there-been-no-great-women-artists
[5] Orientalism (Goodreads editions page): https://www.goodreads.com/book/show/23300798-
    orientalism
[6] The Predicament of Culture (Oxfam Books listing): https://www.oxfambookshop.au/products
    /581532_clifford_predicamentofculture
[7] Global Art History (De Gruyter/Brill): https://www.degruyter.com/document/cover/isbn
    /9783839440612/product_pages
[8] African Indigenous Knowledges in a Postcolonial World (Routledge): https://www.
    routledge.com/African-Indigenous-Knowledges-in-a-Postcolonial-World-A-Tribute-to-Toyin
    /Afolayan-Babalola-Ibrahim/p/book/9780367516833

```
[9] Artforum (letters image associated with McEvilley's critique): https://www.artforum.com
    /wp-content/uploads/2016/09/article00_large-359.jpg
[10] Artforum (additional page image referencing the critique): https://www.artforum.com/wp
     -content/uploads/2017/01/article_large-261.jpg
[11] The Met Heilbrunn Timeline of Art History: https://www.metmuseum.org/toah
[12] Oxford Art Online (Grove Art Online): https://digital.library.wisc.edu/1711.web/
     oxfordart
[13] Smarthistory (About/Essays): https://smarthistory.org/
[14] JSTOR Archive Journals (about access): https://eifl.net/e-resources/jstor-archive-
     journals
[15] Project MUSE (About): https://about.muse.jhu.edu/
[16] Unit 7: Trade and Artistic Exchange (Met educator curriculum, Islamic world): https://
     www.metmuseum.org/learn/educators/curriculum-resources/~/media/Files/Learn/For%20
     Educators/Publications%20for%20Educators/Islamic%20Teacher%20Resource/Unit7.pdf
[17] China, 500-1000 A.D. | Heilbrunn Timeline of Art History: https://www.metmuseum.org/
     toah/ht/06/eac.html
[18] China, 1000-1400 A.D.: https://www.metmuseum.org/toah/ht/07/eac.html
[19] Eastern and Southern Africa, 1400-1600 A.D.: https://www.metmuseum.org/toah/ht/08/afa.
     html
[20] Egypt, 1400-1600 A.D.: https://www.metmuseum.org/toah/ht/08/afe.html
[21] Guinea Coast, 1400-1600 A.D.: https://www.metmuseum.org/toah/ht/08/afg.html
[22] Western and Central Sudan, 1000-1400 A.D.: https://www.metmuseum.org/toah/ht/07/afu.
     html
[23] Age of Transition: Byzantine Culture in the Islamic World (Met symposia volume): https
     ://resources.metmuseum.org/resources/metpublications/pdf/
     Age_of_Transition_Byzantine_Culture_in_the_Islamic_World.pdf
[24] Tile from a Squinch (muqarnas element; Met 20.120.189): https://www.metmuseum.org/art/
     collection/search/447256
[25] Silks from Ottoman Turkey (Bursa sericulture; nakkashane; saz/chintamani): https://www
     .metmuseum.org/essays/silks-from-ottoman-turkey
[26] Woodblock Prints in the Ukiyo-e Style (technique; kento; collaboration): https://www.
     metmuseum.org/essays/woodblock-prints-in-the-ukiyo-e-style
[27] Art of the Pleasure Quarters and the Ukiyo-e Style (urban culture; subjects; markets):
      https://www.metmuseum.org/essays/art-of-the-pleasure-quarters-and-the-ukiyo-e-style
[28] The Print in the Nineteenth Century (industrial prints; etching revival; Japonisme):
     https://www.metmuseum.org/essays/the-print-in-the-nineteenth-century

Notes on sources and images: All images embedded above are drawn exclusively from the
    provided resources. Assertions are restricted to the content of the cited materials.
    Gaps are acknowledged explicitly where information is not provided.
```

Figure 12: Qualitative outputs for the same prompt in Figure 1 from DEERFLOW (a) and DEERFLOW+ (b). Our DEERFLOW+ retains more retrieved information, presents stronger evidence structuring and citation discipline, while avoiding token-limit failures seen in the vanilla version.

## E  EVALUATING CITATION ACCURACY WITH RUBRIC TREE

**Evaluating citation accuracy with a rubric tree.**  In citation-grounded reports, links may be hallucinated, irrelevant to the topic, or fail to substantiate the associated statement. To capture this, we employ an agentic judge that performs a structured validation process: (1) identify each claim with its linked source URL, (2) check whether the URL is accessible, and (3) if accessible, assess whether the content sufficiently supports the claim. The entire process can be visualized as a rubric tree (Figure 13).

**Optimization.**  As reports may contain extensive citations, verifying each link individually would be time-consuming and costly. To improve efficiency, we group all claims associated with the same link, requiring only a single web retrieval for the grouped claims. In addition, before checking whether the content supports a claim, we first perform a coarse-grained relevance check by examining the initial portion of the webpage (*e.g.,* title or first few hundred tokens). For example, if the task concerns healthcare but the link directs to a technology article, we terminate early without further evaluation. In total, we identify three types of errors. E1: URL is inaccessible or does not resolve; E2: URL content is irrelevant to the topic of the task; E3: URL content does not support the specific statements.

**SoTA Deep research systems are far from citation error-free.**  Using our rubric-tree framework, we evaluate the top performers from Table 1 (GPT-5, Grok-4 Deep Research, and Open Deep Research) on the two most search-intensive tasks: market analysis and wide information search. The results are shown in Table 7, where for each agentic system and task category, we report the average number of E1, E2, and E3 errors in the generated reports. We can see that all models produce non-trivial citation errors. In wide information search, most errors stem from *unsupported claims*

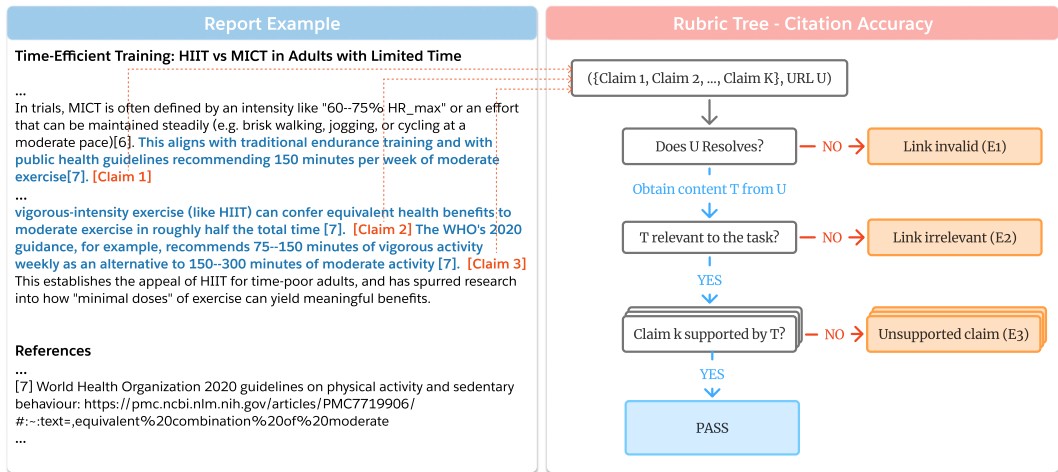

Figure 13: Illustration of rubric tree-based evaluation of citation accuracy. Compared to evaluating all claim-URL pairs individually, the rubric tree groups claims by link, which significantly reduces evaluation costs while enabling finer-grained categorization of citation errors: invalid links (E1), irrelevant links (E2), and unsupported claims (E3).

(claims not verifiable from the cited link) rather than invalid or irrelevant URLs[2], highlighting that hallucinations persist even with web access. The problem is worse for market analysis, where all systems generate large numbers of unsupported claims (Open Deep Research averages 91.9 errors per report), underscoring citation accuracy as a persistent bottleneck in deep research.

| Agent Name | E1 Errors | E2 Errors | E3 Errors | Total |
|---|---|---|---|---|
| **Task: Wide Info Search** | | | | |
| GPT-5 | 4.2 | 1.7 | 13.3 | 19.2 |
| Grok-4 Deep Research | 6.8 | 6.8 | 33.4 | 47.0 |
| Open Deep Research | 5.0 | 5.2 | 19.7 | 29.9 |
| **Task: Market Analysis** | | | | |
| GPT-5 | 11.1 | 10.1 | 43.8 | 65.0 |
| Grok-4 Deep Research | 6.3 | 6.4 | 61.5 | 74.2 |
| Open Deep Research | 11.9 | 11.6 | 68.4 | 91.9 |

Table 7: Evaluation of leading research agents in citation accuracy for search-intensive tasks. For each agentic system, we report the the average number of errors for each task (less is better). We consider three types of errors: URL invalid (E1), URL irrelevant (E2), and Unsupported claim (E3).

# F    LIVERESEARCHBENCH DEMONSTRATION

We provide concrete queries in LiveResearchBench across different task categories in Table 8, Table 9, and Table 10.

# G    ERROR PATTERN EXAMPLES

In this section, we provide concrete examples of error patterns observed in our pilot study. This includes mismatched in-text citations and references (Figure 14), missing links or incomplete URLs (Figure 15), inconsistent citation formats (Figure 16), uncited references appearing in the bibliography (Figure 17), broken or incomplete table formatting (Figure 18), out-of-order reference numbering (Figure 19), embedded citations breaking text flow (Figure 20), and hallucinated information

---

[2]E1 and E2 are counted at the URL level (each problematic URL contributes one error). E3 is counted at the statement level: all citations attached to a claim are expected to support it (*e.g.,* if a claim about vaccination policy cites [5,10,12,24,25], the E3 error count is incremented by one for every source not supporting the claim.

| Category | Query |
|---|---|
| Competitive Analysis | Compare Spotify, Apple Music, Youtube Music, Amazon Music, Pandora, and Tidal on music discovery features (including algorithmic recommendations, curated playlists, and social discovery), audio quality (with reference to both technical specifications and perceived listening experience), and podcast selection (considering global availability), user interface design and ease of navigation, music library size and catalog completeness, exclusive content and early releases, pricing structures for individual and family plans, artist compensation and payout rates, and compatibility with high-end audio equipment as of now. |
| Literature Review | Help me produce a detailed academic report (literature review) on the evolution of evaluation practices for single-agent and multi-agent systems (based on large language models). Focus on the works published from 2023-2025 (including high-quality published papers, technical reports, and preprints). The report should include the following dimensions: (1) Benchmarks and sandbox environments: How do benchmarks for single-agent systems differ from those for multi-agent research in terms of task diversity, scalability, and realism? What are the trade-offs made to create the benchmark, their strengths and limitations? (2) Metrics: What families of evaluation metrics are most frequently applied in single-agent systems versus multi-agent systems, and what trade-offs do they entail? Does the metric rely on LLM-as-judge, agent-as-judge, human, or string matching? (3) Implementation and cost: How is the benchmark/sandbox implemented? Is human annotation involved in creating the benchmark or sandbox and how is human annotation involved (if applicable)? Is the benchmark/sandbox scalable and reliable? (4) Future Directions: What concrete research directions and design principles can advance the development of unified, generalizable evaluation pipelines? |
| Market Analysis | What has been the size, growth rate, and geographic distribution of the COVID-19 vaccine market between 2021 and 2025? What were the key demand phases (initial rollout, booster campaigns, variant-specific vaccines), and how have public health policies and supply constraints influenced vaccine adoption globally? How do Moderna, Pfizer-BioNTech, Johnson & Johnson, and Novavax compare in terms of global dose distribution volumes, clinical efficacy (trial versus real-world data), and manufacturing capacity? What flagship vaccines define their market strategies, and how is the market expected to evolve considering waning demand and shifting public health priorities? |

Table 8: Example queries in LiveResearchBench (Part I).

(Figure 21). These representative cases illustrate the diverse range of structural and citation-related issues that can undermine the clarity and overall reliability of generated research reports. These observations motivated the tailored evaluation metrics in DeepEval as described in Section 4.

| Category | Query |
|---|---|
| Decision Support | In adults aged 30–70 with limited weekly training time ($< 150$ min/week; with a $< 60$ min/week subgroup), what is the comparative effectiveness of high-intensity interval training (HIIT; including low-volume/sprint-interval variants), moderate-intensity continuous training (MICT), and hybrid HIIT+MICT protocols on (a) cardiorespiratory fitness ($VO_2max/VO_2peak$, $mL \cdot kg^{-1} \cdot min^{-1}$), (b) cardiometabolic outcomes (HbA1c/fasting glucose, HOMA-IR, blood pressure, lipids, adiposity/liver fat), and (c) patient-centered outcomes (adherence, enjoyment, adverse events)? How are these protocols defined and evaluated across international randomized trials and meta-analyses (2018–2025), with supportive context from observational studies and major guidelines (e.g., ACSM/WHO), and which progression/maintenance plans (session frequency, intensity prescription, minimal effective dose; supervised vs. home-based delivery) sustain gains over $\geq 6$–12 months? Consider moderators including sex, age band (30–49 vs. 50–70), baseline fitness, and cardiometabolic risk (overweight/obesity, prediabetes/metabolic syndrome, hypertension), as well as medication status. |
| Policy & Regulation | I work on information security and am responsible for incident disclosure, compare CISA's CIRCIA regulations, the SEC's cybersecurity disclosure rules, and sector-specific mandates under NERC CIP. Help me provide a cross-sector analysis (energy, finance, healthcare, and other critical infrastructure), focusing on the most current compliance challenges from the past 1-2 years. Analyze across reporting timelines, covered-entity definitions, enforcement mechanisms, and penalty structures. Provide: - A short description of each regulatory framework - A multi-column comparison table covering all dimensions - Multiple real-world examples of enterprise compliance challenges per framework - An analysis of strategic implications for U.S.-based and global enterprises operating under U.S. jurisdiction across multiple critical infrastructure sectors |
| Pros & Cons | Conduct a comprehensive analysis on whether social media is good for society, with a primary focus on Europe (including the UK, EU member states, and Nordic countries). The analysis should focus specifically on teenagers (ages 13-18) and their relationship with social media. For this report, define "good for society" across four dimensions: 1) mental health and well-being (e.g., anxiety, depression, self-esteem, and loneliness), 2) educational and developmental impacts (e.g., learning, attention span, and digital literacy), 3) social and civil outcomes (e.g., friendships, community engagement, political awareness, and misinformation exposure), and 4) safety and risk factors (e.g., cyberbullying, privacy, exploitation, and screen-time effects). In the report, I would like to have 6 pro arguments and 6 con arguments. Each pro or con argument should include: (1) A clear thesis statement (2) Multiple supporting points substantiated with credible evidence (including but not limited to statistics, direct quotes, case studies, expert testimony, industry or government reports, academic research, news sources, online forums, and social media), followed by in-depth reasoning and analysis. Prioritize data and research from the past 5 years where available (2020-2025). |

Table 9: Example queries in LiveResearchBench (Part II).

| Category | Query |
|---|---|
| Topic Exploration | As someone who created an online photo editing tool targeting social media marketers and small businesses, help me understand how AI is transforming the photo editing space and what I need to do to stay competitive. My tool currently offers basic editing features like cropping, filters, and text overlays, and I'm considering adding AI capabilities through third-party APIs rather than building in-house. I want to know (1) which specific AI features would be most valuable for social media marketers - that I could integrate without massive development costs, (2) whether AI will make traditional photo editing skills obsolete or if there's still a place for manual editing tools alongside AI features, (3) how major players like Canva, Photoshop, and newer AI-first competitors are using AI to attract users, and (4) what the real costs and benefits are for integrating AI APIs into my platform, including monthly API fees, development timeline, server load, and impact on user engagement and premium conversions. Give me examples from photo editing tools serving similar markets, with specific AI features, costs, and results. |
| Wide Info Search | Identify U.S. and Canadian banks and broker-dealers that, according to publicly available sources, launched (i.e., production, not pilot) client-facing AI portfolio-management or financial-planning 'copilots' during 2023-2025, and compile a due-diligence table covering: institution; assistant/product name; launch month; customer tier; geography & languages; core AI stack; safety & compliance; integrations (e.g., custodian/portfolio accounting/CRM/planning tools, SSO, audit logging); access & pricing; KPIs (e.g., adoption, satisfaction, ROI) where available; plus links to the bank/OEM press release, product page or demo, and at least one independent wealth-tech or fintech-trade article confirming the launch. |
| Top Rankings | What are the top 20 gaming monitors released or available for purchase in 2025, based on refresh rate (Hz), response time (ms), and price per inch of screen size? Please assume all three criteria are weighted equally. No specific panel technologies or size requirements are set; consider all currently available monitors within this timeframe. |

Table 10: Example queries in LiveResearchBench (Part III).

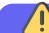
**⚠ Mismatched in-text citations and references**

**Report from Manus:**

**The Electric Truck Market: 2024-2025 Analysis**

**Executive Summary**

The electric truck market is undergoing a transformative period, characterized by rapid growth, evolving technological advancements, and increasing adoption across various customer segments. This report provides a comprehensive analysis of the market's size, demand trajectory, and segmentation for 2024-2025. It delves into the key customer segments driving growth–commercial fleets, consumers, and government–and examines the critical infrastructure and regulatory factors influencing adoption.
...

**1.1 Market Size**

The global electric truck market is experiencing significant growth. Several reports provide varying but consistently upward trends for market valuation:

* **2024:** Projections for the global electric truck market size in 2024 range from USD 1.22 billion [7] to USD 29.69 billion [1, 4, 6] and USD 3.99 billion [8, 14]. The discrepancies likely stem from different methodologies and inclusion criteria (e.g., light-duty vs. medium/heavy-duty, regional vs. global).
* **2025:** The market is projected to grow to USD 4.71 billion [14], USD 5.22 billion [5, 13], or USD 5.88 billion [8] in 2025. Some reports indicate a growth to USD 29.7 billion in 2024 and USD 124.97 billion by 2030 [1, 4, 6].

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

...

**Analysis:**
The report reuses citation numbers across multiple reference lists, causing ambiguity in mapping in-text citations to sources.

Figure 14: Error pattern: mismatched in-text citations and references.

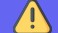

**Missing Links (URLs)**

**Report from GPT-4.1:**

**Clinical Presentation & Differential Diagnoses**

...

**1.1 Extensor Carpi Ulnaris (ECU) Tendon Pathology**
- **ECU Tendinopathy**: Repetitive microtrauma during supination combined with ulnar deviation can cause overuse injury to the ECU tendon, common in racket sports [1][2]. Features include pain on the ulnar side of the wrist, swelling, and discomfort during resisted wrist extension with ulnar deviation—symptomatic of ECU tendinopathy [2][9].
- **ECU Instability or Subluxation**: Sudden onset "bolt out of the blue" pain over the ulnar wrist, especially during double-handed backhand strokes, may indicate traumatic ECU instability (subluxation or dislocation) [1].

...

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

...

**Analysis:**
In-text citations (e.g., [2], [9]) either lack a proper reference entry or appear in the reference list without URLs or identifiable sources.

Figure 15: Error pattern: missing links (URLs).

⚠️ **Inconsistent Citation Format**

**Report from Gemini-2.5 Pro:**

**Major U.S. Bank Received a Less-Than-Satisfactory Community Reinvestment Act Rating in Recent Years**

...

**Washington D.C.** - A comprehensive review of Community Reinvestment Act (CRA) ratings for U.S. banks with assets exceeding $50 billion reveals that only one institution, City National Bank, received a "less-than-satisfactory" rating in its most recent evaluation between 2019 and 2024. The bank was rated "Needs to Improve" by the Office of the Comptroller of the Currency (OCC) in an examination dated March 27, 2023, which covered the years 2019 through 2021.

...

Below are the detailed findings for City National Bank:

**City National Bank**

* **Rating**: Needs to Improve
* **Exam Date**: March 27, 2023 (Evaluation Period: January 1, 2019 - December 31, 2021)
* **Federal Reserve/OCC CRA Disclosure:** [Office of the Comptroller of the Currency (OCC) Performance Evaluation](https://www.occ.gov/static/cra/cra-eval/apr24/14695.pdf)[1](https://www.occ.treas.gov/news-issuances/news-releases/2024/nr-occ-2024-58.html)
* **Reputable News Source:** [City National Bank Press Release](https://www.cnb.com/about-us/newsroom/2024/city-national-bank-community-reinvestment-act-rating-lowered.html)

....

While the vast majority of large U.S. banks maintained "Satisfactory" or "Outstanding" CRA ratings during this period, the case of City National Bank underscores the continued importance of the Community Reinvestment Act in ensuring equitable access to credit for all communities.

**Analysis:**
The report mixes multiple citation styles within the same section. For example, the "Federal Reserve/ OCC CRA Disclosure" entry combines a descriptive hyperlink with a numbered reference [1], while other entries only use hyperlinks.

Figure 16: Error pattern: inconsistent citation format.

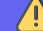

**⚠ References Not Mentioned in Report**

**Report from Manus:**

**U.S. Electric Vehicle Market Analysis**
...
**1.1 Market Size**

In 2024, the U.S. electric vehicle market demonstrated robust expansion in both sales volume and revenue. Full-year EV sales in the United States ranged from approximately 1.3 million to 1.6 million units [1, 3, 5]. This volume translated into an EV sales share of around 10% of all light-duty vehicles sold during the year [1, 3]. In terms of economic value, the U.S. electric vehicle market was estimated to be worth USD 131.3 billion in 2024 [11].

**1.2 Growth Rate**

The market's growth trajectory remained positive, albeit with some variations in reported figures. EV sales in 2024 increased by 7.3% compared to 2023 [5]. The broader U.S. and Canadian market collectively saw a 9% growth in 2024 [9]. Looking ahead, the market is projected to maintain a strong upward trend, with an anticipated annual growth rate (CAGR) of 12.61% between 2025 and 2029, which is expected to result in a market volume of US$154.2 billion by 2029 [7].
...

Adaptation to Pressures: Ford is adapting to competitive and regulatory pressures by broadening its electrification strategy and focusing on competitive battery costs [78]. The company is also navigating evolving emissions and safety standards globally [79]. Ford's price cuts on the Mach-E reflect its adaptation to a competitive market where pricing is a key factor [81].

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

**Analysis:**
The reference list includes entries (e.g., [95]–[98]) that are not cited anywhere in the report body. This creates misalignment between citations and references.

Figure 17: Error pattern: references not mentioned in the report.

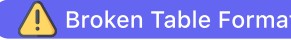

**⚠ Broken Table Format**

**Report from Gemini-2.5 Flash:**

**Comprehensive Research Report: Global ESG Reporting Strategies (2025)**

As a sustainability consultant, understanding the evolving landscape of global ESG reporting is critical for multinational enterprises. This report provides a detailed comparison of the most recent versions of the SEC Climate Disclosure Rule, the EU Corporate Sustainability Reporting Directive (CSRD), and California's climate disclosure legislation (SB-253 and SB-261) as of 2025. It addresses key dimensions, offers insights into compliance challenges and costs, and analyzes strategic implications for companies operating across these jurisdictions.

...

**2. Multi-Column Comparison Table**

| Dimension | SEC Climate Disclosure Rule (as of 2025, currently stayed)| EU Corporate Sustainability Reporting Directive (CSRD) (as of 2025)| California SB-253 & SB-261 (as of 2025)

It's important to note that the regulatory landscape for ESG reporting is dynamic, and some of the rules mentioned, particularly the SEC Climate Disclosure Rule, are facing legal challenges and potential amendments. My analysis is based on the *most recent versions available as of September 2025*, considering reported proposals and statuses.

...

**Analysis:**
The report contains an incomplete comparison table. While the header row is generated, no content rows are provided, leaving only blank space before resuming the main text. This broken formatting disrupts readability and prevents the table from serving its intended comparative function.

Figure 18: Error pattern: broken table format.

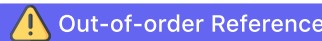

**Report from Manus:**

**Sleep and Body-Clock Strategies for Enhanced Alertness, Learning, and Mood**
...
**Specific Considerations for Shift Work:**
Shift workers face significant challenges in maintaining circadian alignment. Strategies include:
*   **Fixed Shift Schedules:** Where possible, consistent shift schedules (e.g., permanent night shifts rather than rotating shifts) can help the body adapt more effectively [70].
*   **Light Management:** Use bright light during night shifts to promote alertness and wear dark glasses on the commute home to minimize light exposure before daytime sleep [66].
*   **Melatonin Supplementation:** Under medical guidance, melatonin can be used to help adjust to new sleep-wake cycles, particularly when transitioning between shifts or dealing with jet lag [39].

**References (continued)**
[63] Sleep and Academic Excellence: A Deeper Look. (2024). *Stanford Longevity*. [https://longevity.stanford.edu/lifestyle/2024/01/10/sleep-and-academic-excellence-a-deeper-look/](https://longevity.stanford.edu/lifestyle/2024/01/10/sleep-and-academic-excellence-a-deeper-look/)
[64] Polyphasic Sleep: Pros and Cons of Short Sleep Schedule. (2018). *Verywell Health*. [https://www.verywellhealth.com/pros-and-cons-of-a-polyphasic-sleep-schedule-4165843](https://www.verywellhealth.com/pros-and-cons-of-a-polyphasic-sleep-schedule-4165843)
[65] Sleep hygiene: Simple practices for better rest. (2025). *Harvard Health Publishing*. [https://www.health.harvard.edu/staying-healthy/sleep-hygiene-simple-practices-for-better-rest](https://www.health.harvard.edu/staying-healthy/sleep-hygiene-simple-practices-for-better-rest)
...

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

...

**Analysis:**
The reference list is misordered: instead of starting at [1] and proceeding sequentially, it begins at [63]–[65] and then loops back to [1]–[62]. This disrupts the expected citation order and makes it difficult to trace in-text citations accurately.

Figure 19: Error pattern: out-of-order references.

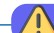 Embedded Citations Breaking Text Flow

**Report from Gemini-2.5 Pro:**

**Mitigating the Mental Toll: A Research Report on Stress and Burnout Prevention for Early-Career Academics**

...

A 2022 study demonstrated that both aerobic and strength training significantly reduced emotional exhaustion, cynicism, and academic inefficacy in university students. A[2](https://vertexaisearch.cloud.google.com/grounding-api-redirect/AUZIYQF9CkyP32RFLyPS3HrW6tR2ye6DRsXqXgt4Qg_uPDWMBLuAMC013grt1GNVouy_JrzwWi_u41df8IsSbZdUqAn15KsjKgx0qhOMjRYykKqZjbEKfyvvUUpL9a3kipVDlv_sflddv84M1MY9mQ==)nother study in 2022 found a strong negative correlation between physical activity and burnout. E[3](https://vertexaisearch.cloud.google.com/grounding-api-redirect/AUZIYQF6hMInhqc9zV_TB5rI3TQ2PBGPVAYPjtBlWtZqaWHxIp1N40ekTwQ-V3Cj3LD_NPPLt_OPeIZSEXzdDukTdXFIh_rb1uHTzgK0axmbEOT2nbTncfEmLvBSfe3OuL9xSQJ2Fng=)xercise not only mitigates the physiological symptoms of stress but also improves self-efficacy and resilience. *[4](https://vertexaisearch.cloud.google.com/grounding-api-redirect/AUZIYQFjuHfGCgNCB3cDIjvukxISoon4g8mnJVgxC4A9hr1sN7af4fi5Jyr9R6FNVHGtRKZgDep9uSfLR-1hg64-zZrN22a9DHYMVvZGqwPg6D9xJLUSCuwQaHB7eWP-rmzyYjLiohI5Sezjz_tbc_c5evuAHdQbSOGmNLMBZDgMsTnJE6JV2QPK_niPqtDvwEI4esLs0j54pbwMLKUoIJ4axgE8v9CGoLjxIlv-fBxQFESHrdGKFYzASamFMzU8pproHHiNxYBTiAkDBPwHIewnIcc4m9RzxQCXyQ7OdKqGnUNVZht7zo355_j9D94PlBXJbOVVv8JKKIjL_K9EE3uEHxDPfbpP57aSjxvsH5ns0WW5Im5YZG-E_A==) **The Strength of Peer and Social Support:** Graduate school and early-career research can be isolating. Formal peer support programs, including peer mentoring and peer-led support groups, have been shown to enhance social connection, reduce loneliness, and provide a sense of community.

...

**Analysis:**
The report disrupts fluency by embedding citations within words (e.g., "A[2]nother," "E[3]xercise"), which breaks the reading flow and appears unedited.

Figure 20: Error pattern: embedded citations breaking text flow.

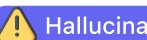 Hallucinated information

**Report from Manus:**

**Top Running Shoes Analysis (August 2025)**
...
## Top Running Shoes Ranking

Based on the analysis, the following running shoes are ranked:

1. **Saucony Endorphin Elite 2**
   * Weight: 0.0 oz
   * Drop: 0.0 mm
   * Runner's World Rating Category: Best Lightweight Racer
   * Composite Score: 1.11
...
6. **Brooks Caldera 8**
   * Weight: 4.5 oz
   * Drop: 0.0 mm
   * Runner's World Rating Category: Best Overall Trail
   * Composite Score: 0.53
...
9. **Lululemon Wildfeel**
   * Weight: 1.0 oz
   * Drop: 7.0 mm
   * Runner's World Rating Category: Best Hybrid
   * Composite Score: 0.33
...

**Analysis:**
The report fabricates data and produces a logically inconsistent ranking, presenting hallucinated information (e.g., implausible weights and scores) that appears plausible at a glance but undermines factual accuracy and composition quality.

Figure 21: Error pattern: hallucinated information.

## H    INFORMATION ON THE ANNOTATION TEAM

For data annotation and verification, we worked with professional annotators from Salesforce. All annotators hold advanced degrees in fields including English Literature, Social Science, Computer Science, and Finance, with 2–6 years of experience in data annotation and analysis. Their expertise spans multi-modal evaluation (text, image, and audio) as well as quality control for advanced annotation pipelines.

## I    USER SURVEY

To inform the design of our deep research benchmark, we conducted an online user survey with participants from diverse backgrounds and occupation. We group participants into three categories: enterprise professionals (including sales, marketing, legal, and software engineering; 27%), researchers (including professors, industrial researchers, postdocs, and Ph.D. students; 24%), and general users (the rest; 49%). Participants were asked: `What questions would you ask a deep research agent?`, and `What aspects of the model's response matter most to you?`
We analyzed the collected responses and observed the following trends: (1) many questions are multi-faceted and require real-time search; (2) users value responses that capture both the surface-level request and the implicit intention of the query; and (3) they expect the final report to be tailored to the target audience, in content, format, and presentation style.
For the collected questions, we distilled them into ten representative task categories (Figure 2). Among these, the three most common were *topic understanding* (24.5%), *wide information search* (22.8%), and *top rankings* (16.7%), followed by *market analysis* (8.2%), *technical support* (6.5%), *decision support* (5.2%), *policy & regulation* (4.8%), *literature review* (4.3%), *topic exploration* (3.5%), *pros & cons* (2.0%), and *competitive analysis* (1.5%). In addition, we also analyzed the domain distributions based on the collected user responses: Science and Technology, Economy and Business, Health and Wellbeing, Law and Governance, Society and Culture, Education and Knowledge, Media and Entertainment. Together, these findings were instrumental in shaping the design principles and the curation of tasks in LiveResearchBench.

## J    HUMAN VERIFICATION GUIDELINES

**Goal.**    Verify the quality of deep research queries and their corresponding checklists that aims to assess the comprehensiveness and coverage of the report.

**Data Format**    The CSV contains one row per grading criterion, with the following key columns:

| Column | Description |
| --- | --- |
| qid | Query identifier. |
| Deep Research Query | The detailed deep research query. |
| Checklist | Individual grading questions (Multiple checklist items per qid). |

Table 11: Key columns in the annotation CSV format.

You will be evaluating two components: (1) Deep Research Query Quality and (2) Checklist Quality.

### J.1    ANNOTATION TASKS

#### J.1.1    TASK 1: DEEP RESEARCH QUERY QUALITY

Rate each unique `Deep Research Query` according to Table 12:

| Rating | Description |
| --- | --- |
| **Appropriate** | Query is well-structured, complete, clear, and answerable through research. |
| **Appropriate with modification** | Good foundation but needs improvement (please provide a modified version). |
| **Not appropriate** | Fundamentally flawed or not worth fixing. |

Table 12: Rubric for rating deep research query quality.

Consider the following criteria:

- **Answerability**: Can this question be reasonably answered using web search and analysis?
- **Clarity**: Is it clear what information is being requested? Useful clarifiers include but not limited to temporal bounds, geographic scope, and specific entities.
- **Completeness**: Is the query complete? Would you, as the user, add or remove specifics to make it more complete?
- **format**: When appropriate, is the format and report organization requirement clear? This criterion may not apply to all questionsâĂŤuse it judiciously.
- **Target audience**: When appropriate, is the query framed appropriately for its intended readers (*e.g.*, technical experts, policymakers, or general public)? Does the wording match the expected background knowledge and level of detail of that audience? This criterion may not apply to all questionsâĂŤuse it judiciously.

### J.1.2 TASK 2: GRADING CRITERIA QUALITY

Rate each `Checklist` item according to Table 13:

| Rating | Description |
|---|---|
| Appropriate | Relevant, verifiable, and important. |
| Valid but not necessary | Correct but redundant or overly granular (please comment why; modify if needed). |
| Needs modification | Poorly phrased or too narrow (please provide a modified version). |
| Not appropriate | Irrelevant, unverifiable, or impossible (please comment why; modify if needed). |

Table 13: Rubric for rating checklist quality.

**Remark:** If additional checklist items should be included, you should append the new checklist items for the corresponding qid.

When evaluating each checklist item, consider the following criteria:

- **Relevance**: Does the criterion correspond to the requirements of the research question?
- **Verifiability**: Can it be objectively verified from a research report?
- **Importance**: Does it capture a meaningful aspect of the research question?
- **Non-redundancy**: Does it avoid substantial overlap with other checklist items?

### J.2 CONCRETE EXAMPLES

### J.2.1 TASK 1: DEEP RESEARCH QUERY

> **RQ1**: What is the size, growth rate, and segmentation of the U.S. electric vehicle market in 2025? What are the key drivers (policy incentives, charging infrastructure, consumer adoption) and challenges (supply chain, cost pressures)? How do Tesla, BYD, Volkswagen's ID series, and Ford Mustang Mach-E compare in market share, production capacity, and pricing philosophies across major regions?

**Rating: Appropriate.** The query is well-structured with clear sub-questions, specifies a concrete temporal scope (2024) and geographic focus (U.S.), mentions relevant entities (Tesla, BYD, etc.), and is answerable through standard market research. Note that the query is well scoped with with a business/market research context, and it doesn't hinge on whether the target audience is, say, a policymaker, an investor, or the general public—all of those groups could use essentially the same data. Therefore, there is no need to further specify the target audience in this case.

> **RQ2**: How has LLM technology evolved and what are the current trends?

**Rating: Needs Modification or Inappropriate.** The query is overly broad ("LLM technology" is unspecified), lacks temporal bounds, and sets vague expectations ("current trends" without context).

In addition, it does not reflect the needs of a specific target audience—while researchers may require technical detail, the general phrasing here fails to indicate whether the query is meant for experts, practitioners, or the broader public.

### J.2.2 TASK 2: CHECKLIST ITEMS

> **C1**: Does the report provide data on the electric vehicle market specifically in U.S.?

**Rating: Appropriate.** This checklist item is directly relevant to the research question, as it captures the query's explicit focus on the U.S. electric vehicle market. It is objectively verifiable, since one can check whether the report includes U.S.-specific market data. It addresses an important dimension of the query by ensuring geographic precision, and it does so without overlapping with other checklist items.

> **C2**: Does the report mention Tesla by name?

**Rating: Valid but not necessary.** This checklist item is correct and verifiable, since Tesla is explicitly part of the query. However, it is overly granular: focusing only on Tesla misses the broader comparison with other key entities such as BYD, Volkswagen, and Ford.
**Comment:** Redundant if there is already a broader criterion covering multiple entities.

> **C3**: Does the report identify ALL electric vehicle companies in the market?

**Rating: Not appropriate.** This checklist item is unverifiable in practice, as there is no authoritative ground truth for identifying *all* electric vehicle companies. It also sets an impossible standard, since completeness across the entire market cannot reasonably be expected from a single report. A more suitable criterion would focus on specific entities mentioned in the query (e.g., Tesla, BYD, Volkswagen, Ford) rather than requiring exhaustive coverage.

### J.3 REMARKS AND TIPS

Each deep research query should be rated once for its corresponding qid, while each checklist item should be rated individually. Several tips for annotation: (1) avoid checklist items that demand unverifiable completeness, such as those asking for "all" or "every" entity or fact; (2) watch for redundancy, where multiple criteria check essentially the same requirement; (3) be cautious of scope creep, in which a checklist item evaluates something not actually specified in the query; (4) exclude subjective judgments; checklist items should focus on verifiable facts rather than opinions.

## K RESULTS BREAKDOWN

Figure 22 reports the breakdown of the ten presentation checklist items (Table 2) for each system. High-level organization is largely under control: models perform well on P1 (section-level structure), and straightforward behaviors such as placing citations at clause/sentence boundaries (P7) and ensuring figures/tables contain valid entries (P8) are near-saturated. The harder cases are the mechanics of style and bibliographic control: *style/formatting* (P2, P5–P6, P9) and *citation management* (P3–P4, P10) show wide variance across systems. We also observe provider-level regularities: sibling models tend to share strengths and weaknesses (e.g., on P6, GPT-5 / GPT-4.1 / GPT-5 Mini score 31%, 32%, 24%), likely due to shared data and formatting designs. System design can shift these profiles: multi-agent wrappers around a common backbone often improve P4 by enforcing explicit aggregation/mapping from in-text citations to references, whereas numbering accuracy in P10 varies by pipeline, with some single-agent web systems exhibiting steadier numbering. Overall, a consistent pattern emerges across families: *high-level coherence and simple placement/validity are easy; citation correctness, style uniformity, grammar, and global formatting discipline remain the hard parts*.

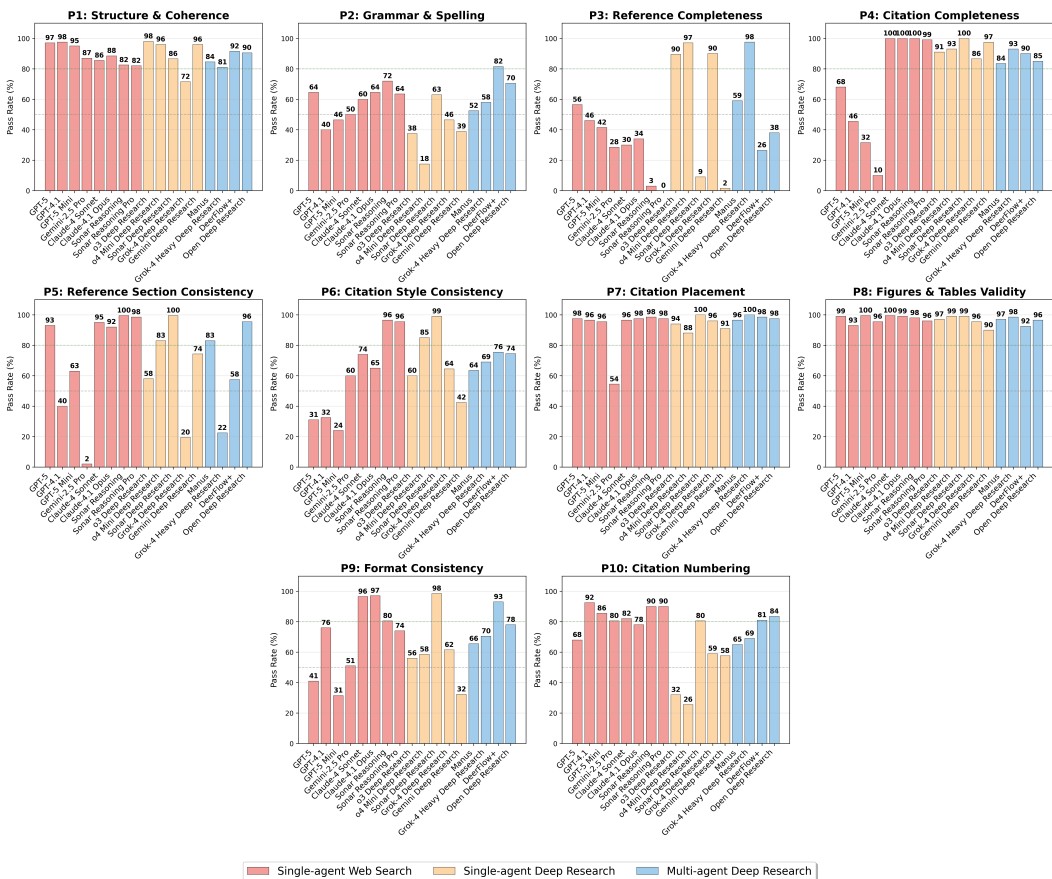

Figure 22: Fine-grained pass rates (P1–P10) for the *Presentation & Organization* metric. Most systems are reliable on P1 (Structure & Coherence) and near-saturated on P7 (Citation Placement) and P8 (Figures & Tables Validity). The largest dispersion appears in grammar, citation style/formatting, and reference alignment (P2–P6, P9, P10). Models from the same provider exhibit correlated patterns; pipeline design (e.g., multi-agent wrappers) can help citation-focused tests.

