# OpenReview forum: "LiveResearchBench: A Live Benchmark for User-Centric Deep Research in the Wild"
_ICLR.cc/2026/Conference — ICLR 2026 Poster_

### Official Review · Reviewer_Mkd1 · 2025-10-18

**Soundness:** 3
**Presentation:** 3
**Contribution:** 3
**Rating:** 6
**Confidence:** 4

**Summary:**

This paper presents LiveResearchBench, a new 100-task benchmark for "deep research" agents, and an evaluation suite, DeepEval. The authors use this framework to test 18 different systems, offering some useful insights into their common failure points.
Overall, this is a valuable and thoughtfully designed contribution to the field.

**Strengths:**

The paper’s main strength is its design philosophy. The authors do an excellent job motivating why this kind of evaluation needs to be user-centric, dynamic, and unambiguous, and they've built the tasks accordingly. The benchmark itself feels realistic, with tasks that specify an audience and output format while requiring up-to-date information, which is a convincing improvement over prior work.
The breadth of the DeepEval suite is also good. It combines high-level checks with more granular metrics, like checklist-based coverage and a rubric-tree for citation validation, which seem to align well with human judgment. Evaluating 18 systems provides a solid snapshot of the current landscape and surfaces recurring errors that will be helpful for the community.

**Weaknesses:**

My main issue is the evaluation pipeline's heavy reliance on LLMs for generating checklists and judging outputs. While the authors took steps to mitigate bias, it's still a risk. I'd be more convinced by more quantitative stats on model-human agreement or some ablation studies on the judging process.

**Questions:**

1. How do you define `wide information search`? Any idea related to the existing Wide Research?
2. How did you handle potentially harmful queries or unreliable web sources during the evaluation?

---

> ### Author Response · Authors · 2025-11-21
> **Response to Reviewer Mkd1 (1/3)**
>
> Thank you for your insightful feedback, suggestions, and support of our work! In the following, we address each comment and question in more detail:
>
> > W: My main issue is the evaluation pipeline's heavy reliance on LLMs for generating checklists and judging outputs. While the authors took steps to mitigate bias, it's still a risk. I'd be more convinced by more quantitative stats on model-human agreement or some ablation studies on the judging process.
>
> Thanks for bringing this up! We are happy to provide further details and our rationales regarding the design choice.
>
> **Checklist construction**. We have ran a pilot study comparing human-written checklists with LLM-generated ones and found that state-of-the-art LLMs such as GPT-5 can reliably produce strong initial drafts (e.g., "Does each listed launch fall strictly between 2023 and 2025?"). However, we **do not** use these drafts directly. Every checklist item undergoes a multi-round human refinement process (five-stage verification pipeline; Fig. 4). During the process, humans will decide to **rewrite, modify, or remove** the checklist if it does not satisfy the criteria. After the final verification stage where a separate group of experts cross-checks every item, the human alignment rate of remaining checklist items is 100%, as any items that do not fully align with human expectations are removed. Thus, despite the initial draft from LLMs, the final benchmark contains only human-approved or human-written checklists.
>
> Our evaluation setup is designed to be reliable and highly aligned with human judgment. The selection of the LLM-ensemble-as-a-judge (Gemini-2.5-Pro + GPT-5) and the design of each dimension-specific protocol are guided by both **run-to-run stability** and **high alignment with human experts**.
> - Ranking Stability: For each model and evaluation dimension, we conducted 3 independent runs. Rankings remained unchanged across all runs.
> - Judge Ensemble Selection: We compare Gemini-2.5-Pro, GPT-5, and Claude-4-Sonnet. The Gemini + GPT-5 ensemble consistently achieves the highest alignment with human judgments and helps mitigate single-model bias.
>
> We have included the detailed statistics and alignment studies in the revised manuscript and provide an overview below:
> - Presentation & Organization:
>     - Human study shows 98.3% agreement with either Gemini or GPT-5
> - Coverage & Comprehensiveness:
>     - Human 100.0% prefers Gemini or GPT-5 as judge; no instance uniquely favored Claude
> - Analysis Depth: Human agreement with Gemini or GPT-5: 93.75%
> - Factual & Logical Consistency and Citation Traceability:
>     - Additive (issue-based) scoring shows better human alignment and yields more stable ratings than direct scoring
>     - Factual & Logical Consistency: for identified issues, the human agreement rate is 82.0%
>     - Citation Traceability: for identified issues, the human agreement rate is 85.9%
>     - Note: Our scoring penalizes every two issues, so final score alignment is even higher.
> - Citation Accuracy: Human agreement across flagged issues: 87.1%

---

> ### Author Response · Authors · 2025-11-21
> **Response to Reviewer Mkd1 (2/3)**
>
> We also provide a more detailed discussion on how previous benchmarks evaluate long-form open-ended reports and how our design addresses their limitations:
> - **DeepResearch Bench** uses a single LLM call (Gemini-2.5-Flash) to assign scores for four dimensions: comprehensiveness, depth, instruction-following, and readability. These scores are then averaged, with the weights also predicted by another LLM call.
> However, in our pilot study, we found that relying on a single LLM to assign holistic scores can lead to **extremely high variance**. For instance, two independent runs using frontier models (e.g., GPT-5, Gemini-2.5-Pro, Claude-4-Opus, Claude-4-Sonnet) can produce performance differences of **over 50 points** on analysis depth, making the results unreliable.
> To address this, we instead use LLM-ensemble-as-a-judge, along with checklist-based, additive, pairwise, and rubric-tree based protocols designed to reduce variance and improve interpretability for each dimension.
> - **ResearcherBench** also uses a single LLM to judge coverage, faithfulness, and groundedness, which correspond roughly to our metrics of Coverage & Comprehensiveness, Citation Traceability, and Citation Accuracy. However, we go further by evaluating three additional dimensions:
>     - Presentation & Organization: Captures how clearly a report communicates its findings, including structural flow, formatting, and citation hygiene. Poor presentation increases cognitive load, makes reasoning harder to follow, and can obscure valid evidence or mislead readers—directly impacting interpretability, traceability, and overall reliability.
>     - Analysis Depth: We hope to evaluate the models research ability beyond pure search and information retrieval.
>     - Factual & Logical Consistency: evaluates internal coherence across the full report, not just localized factual correctness.
>     - These additional dimensions help prevent superficial or easily gaming behavior, such as producing short, contrastive yet well-cited summaries that appear locally coherent but lack analytical depth or global consistency. We also found that omitting an explicit presentation dimension often causes LLM judges to conflate style issues with consistency or citation errors. Isolating it improves diagnostic clarity.
> - **DeepScholar-Bench** primarily targets the *related work section* of academic papers, whereas our benchmark focuses on **long, open-ended research reports across diverse domains**.
>     - Their evaluation uses a single model (GPT-4.1 or GPT-4o) across all dimensions, with no human-verified rubrics. In contrast, we use a LLM ensemble with human-validated checklists.
>     - For Organization, they report a 78% human–LLM agreement rate via pairwise comparisons with reference sections. Our checklist-based method achieves 98.3% agreement.
>     - For Coverage, their rubric is not validated, and the agreement is 72%, while ours achieves 100%.
>     - Moreover, they do not evaluate Analysis Depth, Factual & Logical Consistency, or Citation Traceability, which are central to our evaluation protocol.
>
> This expanded comparison and discussion has been included in Appendix B.
>
>
>
> > Q1: How do you define wide information search? Any idea related to the existing Wide Research?
>
> Wide information search refers to the task that requires extensive web search to discover entities or information that satisfy multiple constraints, followed by synthesizing the findings into a coherent, well-structured report. For example:
>
> > Write a report identifying automakers that launched a generative‑AI voice assistant in production vehicles (public rollout, not pilot) between 2023‑2025? The report should include information (whenever possible) on automaker, assistant name, LLM partner, launch region, launch month, and links to the OEM press release and an automotive‑tech publication. Focus on only passenger vehicle automakers (e.g., no trucks, motorcycles, or commercial fleets).
>
> Such tasks require cross-checking many heterogeneous sources, and presents challenges on long context handling and memory management for agents. In LiveResearchBench, the queries also involve time-sensitive constraints that require real-time search. Our task design shares similar motivations and principles as in recent works on wide search [1], where "a financial analyst may need to find all companies in a sector that meet specific revenue and growth criteria; a job seeker may need to find all job vacancies that match their criteria for role." However, we do not constrain models to fill a predefined table schema as in [1], but instead focus on end-to-end report generation with citation grounding. If helpful, we could provide a clear definition in the revised manuscript and discuss related works in this area.
>
> [1] Wong et al., WideSearch: Benchmarking Agentic Broad Info-Seeking, https://arxiv.org/pdf/2508.07999

---

> ### Author Response · Authors · 2025-11-21
> **Response to Reviewer Mkd1 (3/3)**
>
> > Q2: How did you handle potentially harmful queries or unreliable web sources during the evaluation?
>
> Thank you for raising this important point!
> **Potentially harmful queries**: As all queries in LiveResearchBench are written by humans and pass multiple rounds of review by professional annotators, so none of the queries introduce harmful intent by design.
>
> **Unreliable web sources**: we use a rubric-tree evaluator (Appendix D; Figure 16). Given a set of claims associated with a URL U, the evaluator (1) retrieves the webpage (if the URL resolves), (2) checks whether each claim is supported, and (3) only if the claim is supported, proceeds to validate the quality of the source before marking the claim–URL pair as "PASS".
>  In early experiments, we considered a more detailed categorization of source quality where we divide the quality of sources into three classes: high quality sources (e.g., authoritative and professionally maintained sources), medium-quality sources (e.g., generally reliable but may have lighter editorial oversight or commercial incentives), and low-quality sources. However, we found that these fine distinctions were often subjective and varied by task.
> As the rubric-tree framework is modular, it can be easily extended to incorporate finer-grained source categorization for scenarios where such distinctions are well-defined.

---

### Official Review · Reviewer_wzms · 2025-10-31

**Soundness:** 3
**Presentation:** 3
**Contribution:** 2
**Rating:** 4
**Confidence:** 3

**Summary:**

This paper introduces LiveResearchBench, a benchmark designed to evaluate deep research agents that produce comprehensive, citation-grounded reports through multi-source web investigation. The benchmark consists of 100 expert-curated, realistic, and dynamically updating tasks across domains such as daily life, enterprise, and academia, constructed under three principles: user-centricity, dynamism, and unambiguity. Alongside it, the authors propose DeepEval, a multifaceted evaluation suite assessing report quality through coverage, presentation, citation accuracy, and analytical depth. Empirical evaluation across single- and multi-agent setups highlights limitations in current deep research capabilities.

**Strengths:**

1. The experiment is extensive, including 17 deep research agents' performance on the proposed benchmark.
2. The finding that existing deep research agents struggle with citation reliability and analytical depth is useful.
3. The checklist design is interesting.

**Weaknesses:**

1. The novelty is not enough and the description of related work is not precise. E.g., Mind2Web2 is ambiguous and focuses on short answers, etc. This makes the distinction from the existing work not convincing.

**Questions:**

1. The clearly defined specific questions are not practical. In reality, are people always asking such specific questions?

---

> ### Author Response · Authors · 2025-11-21
> **Response to Reviewer wzms (1/2)**
>
> Thank you for appreciating our extensive experiments, findings useful, and our design interesting! We have addressed your concerns point by point below:
>
> >W: The novelty is not enough and the description of related work is not precise. E.g., Mind2Web2 is ambiguous and focuses on short answers, etc. This makes the distinction from the existing work not convincing.
>
> Thank you for the valuable feedback! In the original submission, our discussion of related work was condensed due to space constraints. We appreciate the opportunity to clarify these comparisons in more detail and have included this expanded discussion in Appendix B. Below, we highlight how LiveResearchBench differs from prior benchmarks across key dimensions:
>
> |Benchmark|Human-verified rubrics/answers|Expert-curated tasks|Open-ended Tasks|Multi-domain|Dynamic Queries|LLM-ensemble-as-a-judge|
> |-|-|-|-|-|-|-|
> |ScholarSearch [1]|✗|✗|✗|✓|✗|✗|
> |BrowseComp [2]|✓|✗|✗|✓|✗|✗|
> |ResearcherBench[3]|✓|✓|✓|✗|✗|✗|
> |DeepScholar-Bench [4]|✗|✗|✓|✗|✓|✗|
> |ReportBench [5]|✗|✗|✗|✓|✓|✗|
> |DeepResearch Bench [6]|✗|✓|✓|✓|✗|✗|
> |Mind2Web2 [7]|✓|✓|✗|✓|✗|✗|
> |LiveDRBench [8]|✗|✗|✗|✓|✗|✗|
> |Deep Research Bench [9]|✓|✓|✗|✓|✗|✗|
> |DeepResearchGym [10]|✗|✗|✗|✓|✗|✗|
> |**LiveResearchBench (Ours)**|**✓**|**✓**|**✓**|**✓**|**✓**|**✓**|
>
> We also now provide a more detailed discussion on how previous benchmarks evaluate long-form outputs and how our design addresses their limitations:
>
> - **DeepResearch Bench** uses a single LLM call (Gemini-2.5-Flash) to assign scores for four dimensions: comprehensiveness, depth, instruction-following, and readability. These scores are then averaged, with the weights also predicted by another LLM call.
> However, in our pilot study, we found that relying on a single LLM to assign holistic scores can lead to **extremely high variance**. For instance, two independent runs using frontier models (e.g., GPT-5, Gemini-2.5-Pro, Claude-4-Opus, Claude-4-Sonnet) can produce performance differences of **over 50 points** on analysis depth, making the results unreliable.
> To address this, we instead use LLM-ensemble-as-a-judge, along with checklist-based, additive, pairwise, and rubric-tree based protocols designed to reduce variance and improve interpretability for each dimension.
> - **ResearcherBench** also uses a single LLM to judge coverage, faithfulness, and groundedness, which correspond roughly to our metrics of Coverage & Comprehensiveness, Citation Traceability, and Citation Accuracy. However, we go further by evaluating three additional dimensions:
>     - Presentation & Organization: Captures how clearly a report communicates its findings, including structural flow, formatting, and citation hygiene. Poor presentation increases cognitive load, makes reasoning harder to follow, and can obscure valid evidence or mislead readers—directly impacting interpretability, traceability, and overall reliability.
>     - Analysis Depth: We hope to evaluate the models research ability beyond pure search and information retrieval.
>     - Factual & Logical Consistency: evaluates internal coherence across the full report, not just localized factual correctness.
>     - These additional dimensions help **prevent superficial or easily gaming behavior**, such as producing short, contrastive yet well-cited summaries that appear locally coherent but lack analytical depth or global consistency. We also found that omitting an explicit presentation dimension often causes LLM judges to conflate style issues with consistency or citation errors. Isolating it improves diagnostic clarity.
> - **DeepScholar-Bench** primarily targets the *related work section* of academic papers, whereas our benchmark focuses on **long, open-ended research reports across diverse domains**.
>     - Their evaluation uses a single model (GPT-4.1 or GPT-4o) across all dimensions, with no human-verified rubrics. In contrast, we use a LLM ensemble with human-validated checklists.
>     - For Organization, they report a 78% human–LLM agreement rate via pairwise comparisons with reference sections. Our checklist-based method achieves 98.3% agreement.
>     - For Coverage, their rubric is not validated by human experts, and the agreement is 72%, while ours achieves 100%.
>     - Moreover, they do not evaluate Analysis Depth, Factual & Logical Consistency, or Citation Traceability, which are central to our evaluation protocol.
>
> [1] https://arxiv.org/abs/2506.13784
>
> [2] https://arxiv.org/abs/2504.12516
>
> [3] https://arxiv.org/abs/2507.16280
>
> [4] https://arxiv.org/abs/2508.20033
>
> [5] https://arxiv.org/abs/2508.15804
>
> [6] https://arxiv.org/abs/2506.11763
>
> [7] https://arxiv.org/abs/2506.21506
>
> [8] https://arxiv.org/abs/2508.04183
>
> [9] https://arxiv.org/abs/2506.06287
>
> [10] https://arxiv.org/abs/2505.19253

---

> ### Author Response · Authors · 2025-11-21
> **Response to Reviewer wzms (2/2)**
>
> > Q: The clearly defined specific questions are not practical. In reality, are people always asking such specific questions?
>
> Thank you for raising this important point! While it is true that the *initial* user queries may indeed be ambiguous, frontier systems (e.g., Gemini Deep Research, OpenAI Deep Research [11]) typically perform **clarification turns** by either asking users to specify the scope or autonomously narrowing the topic before initiating the actual research process. Our tasks are thus based on real queries observed in practical use of deep research systems, and are designed to reflect these **realistic final research queries**: the clarified queries that systems work with after one or more refinement steps. These are the effective prompts used to trigger actual research behavior in deployed systems.
>
> By capturing this final clarified query, we ensure that:
> - All systems receive the same research scope
> - The same system interprets the query consistently across multiple rollouts
> - Generated reports are comparable, enabling fair and rigorous evaluation
>
> Without this design choice, directly using ambiguous prompts would cause the *same system to produce drastically divergent reports across runs*, since its interpretation of the query vary from run to run. This would make system-level comparison unreliable and evaluation results inconsistent.
>
> We hope these address your concerns and we are happy to answer any questions you may have!
>
> [11] https://platform.openai.com/docs/guides/deep-research

---

### Official Review · Reviewer_M3tr · 2025-11-01

**Soundness:** 3
**Presentation:** 2
**Contribution:** 2
**Rating:** 4
**Confidence:** 5

**Summary:**

This paper proposes **LiveResearchBench**, a benchmark for evaluating *Deep Research* agents. It comprises **100 expert-curated, time-sensitive, multi-faceted tasks**, each specified with a clear audience, scope, and expected output to minimize ambiguity. The authors also present a comprehensive evaluation suite (DeepEval) covering six dimensions that span both report-level and content-level assessment. The paper evaluates 18 agent systems and reports a range of observations about the current agents.

**Strengths:**

- While there are a surge of new deep research benchmarks, this work remains different from vey recent works (e.g., unambiguous and time varying research tasks).

- The paper includes a user study and human-agreement check for parts of the evaluation pipeline. These are important for the benchmark design as well as the reliability of the evaluation.

- Comprehensive evaluation on most existing frontier systems.

**Weaknesses:**

Some high-level points:

- Although the benchmark aims to evaluate open-ended research and report generation, the tasks still appear biased to tasks that are over-prescribed (explicit audience, formatting requirements, and tight scopes). This is understandable for the convenience of the evaluation, but still undermines its claimed scope.
- The evaluation is central to the contribution, yet the paper does not clearly explain how each of the six metrics was designed, what prior work they extend, and why these particular choices were made.
- The evaluation heavily depends on LLM-as-a-Judge, which could make it both expensive and unreliable. Given this, the details of human agreement studies should be further detailed in the paper.
- The paper defines six metrics but the main table averages only on four of them? Why omit **Citation Accuracy** in the main comparison when accurate grounding is arguably the linchpin for user need? This is particularly concerning given the text itself notes that models struggle most with **citation correctness/formatting**. Even more worrying, the method that tops the main table (Open Deep Research) is reported to have extremely high overall error rates on some tasks

Regarding the individual metrics:

- Report-presentation metric may have overweighted style and citation formatting (and actually many of them are directly related to citation formats). If the presentation score is driven by citation entries, duplicates, or formatting, it's very likely an agent can overfit to style easily. Might as well elaborate more about how the 10 error patterns were selected and discuss the risk of RL/finetuning leading to superficial gains on these patterns.
- Factual & Logical Consistency at scale. Longer documents are more likely to contain contradictions, but they also become harder for an LLM judge to evaluate reliably. This may be a challenge for the judge model.
- Analysis Depth: Looking at the prompt, it looks too heavy for the LMs to reason accurately over all the points.
- Citation-related two:
  - It is **unclear how claims are extracted** from long reports and how accurate this extraction is (likely another LLM call; but now the writing in the main text, Appendix D, the figure are not clear enough). This step is crucial and should be evaluated itself given that the extraction could have been blocked by the websites.
  - The **rubric-tree framing** of citation accuracy feels unnecessary for what is essentially a simple three-step pipeline

Writing:

- Writing should be further polished. Some details are hard to comprehend and some details in the appendix appear to be made in a rush. The references are also loose and arbirtrary.
- In the relatde work section, the "Evaluation of Long-form Answer" part should discuss more about how previous (search-related) benchmarks handled long-form answers.

**Questions:**

- (Some questions are already mentioned in the weakness section)
- The paper mentions a very large time investment (1500 hours), but it is not clear what is included (task ideation, rubric creation, judge prompts, code, pilot runs, re-annotation, etc.). A breakdown would help much to understand the number.
- DeepEval appears to involve many LLM calls. What is the end-to-end cost and latency to evaluate a single system on the full benchmark?
- Will the benchmark be fully opensourced? (including the tasks, task-specific checklists, code, etc.)
- I'd suggest including a summary table (ideally in the main paper) contrasting LiveResearchBench with representative benchmarks across key dimensions. The current writing is not clear enough and will be hard for people to comprehend.
- OpenAI Deep Research seems more like a multi-agent system, given the information revealed?

---

> ### Author Response · Authors · 2025-11-21
> **Response to Reviewer M3tr (1/5)**
>
> Thank you for your time and insightful review of our work! We greatly appreciate your recognition of the benchmark's distinctiveness, our incorporation of human–LLM agreement studies, and the comprehensiveness of the evaluation! We address your comments point by point below:
>
> > W1: Tasks are over-prescribed. This is understandable for the convenience of the evaluation, but still undermines its claimed scope.
>
> Thank you for raising this important point! Our tasks are based on real queries observed in practical use of deep research systems, and are designed to reflect **realistic final research queries** issued to deep research systems. While the *initial* user queries may indeed be ambiguous, frontier systems (e.g., Gemini Deep Research, OpenAI Deep Research [1]) typically perform **clarification turns** by either asking users to specify the scope or autonomously narrowing the topic before starting the main research process.
>
> Our benchmark therefore captures this *final clarified query*, enabling that:
> - All systems receive the same research scope
> - The same system interprets the query consistently across different rollouts
> - Generated reports are comparable and evaluation remains fair and rigorous
>
> Without this design choice, directly using ambiguous prompts would cause the *same system to produce drastically divergent reports across rollouts*, since its interpretation of the query vary from run to run. This would make system-level comparison unreliable and lead to inconsistent evaluation results.
>
> [1] https://platform.openai.com/docs/guides/deep-research
>
> > W2&Q4&W10: A summary table of comparison to prior benchmarks and evaluation methods
>
> Thank you for the valuable feedback! We have added a summary table that compares LiveResearchBench with representative prior benchmarks across key dimensions. This is now included in the revised main paper.
>
> |Benchmark|Human-verified rubrics/answers|Expert-curated tasks|Open-ended Tasks|Multi-domain|Dynamic Queries|LLM-ensemble-as-a-judge|
> |-|-|-|-|-|-|-|
> |ScholarSearch [2]|✗|✗|✗|✓|✗|✗|
> |BrowseComp [3]|✓|✗|✗|✓|✗|✗|
> |ResearcherBench[4]|✓|✓|✓|✗|✗|✗|
> |DeepScholar-Bench [5]|✗|✗|✓|✗|✓|✗|
> |ReportBench [6]|✗|✗|✗|✓|✓|✗|
> |DeepResearch Bench [7]|✗|✓|✓|✓|✗|✗|
> |Mind2Web2 [8]|✓|✓|✗|✓|✗|✗|
> |LiveDRBench [9]|✗|✗|✗|✓|✗|✗|
> |Deep Research Bench [10]|✓|✓|✗|✓|✗|✗|
> |DeepResearchGym [11]|✗|✗|✗|✓|✗|✗|
> |**LiveResearchBench (Ours)**|**✓**|**✓**|**✓**|**✓**|**✓**|**✓**|
>
> In addition, we now provide a detailed discussion of how prior benchmarks evaluate long-form open-ended outputs, and how our design addresses their limitations:
>
> - **DeepResearch Bench** uses a single LLM call (Gemini-2.5-Flash) to assign scores for four dimensions: comprehensiveness, depth, instruction-following, and readability. These scores are then averaged, with the weights also predicted by another LLM call.
> However, in our pilot study, we found that relying on a single LLM to assign holistic scores can lead to **extremely high variance**. For instance, two independent runs using frontier models (e.g., GPT-5, Gemini-2.5-Pro, Claude-4-Opus, Claude-4-Sonnet) can produce performance differences of over 50% on analysis depth, making the results unreliable.
> To address this, we instead use LLM-ensemble-as-a-judge, along with checklist-based, additive, pairwise, and rubric-tree based protocols designed to reduce variance and improve interpretability for each dimension.
> - **ResearcherBench** also uses a single LLM to judge coverage, faithfulness, and groundedness, which correspond roughly to our metrics of Coverage & Comprehensiveness, Citation Traceability, and Citation Accuracy. However, we go further by evaluating three additional dimensions:
>     - Presentation & Organization: Captures how clearly a report communicates its findings, including structural flow, formatting, and citation hygiene. Poor presentation increases cognitive load, makes reasoning harder to follow, and can obscure valid evidence or mislead readers, directly impacting interpretability, traceability, and overall reliability.
>     - Analysis Depth: We hope to evaluate the models research ability beyond pure search and information retrieval.
>     - Factual & Logical Consistency: evaluates internal coherence across the full report, not just localized factual correctness.
>     - These additional dimensions help *prevent superficial or easily gaming behavior*, such as producing short, contrastive yet well-cited summaries that appear locally coherent but lack analytical depth or global consistency. We also observed that when presentation is not evaluated independently, LLM judges may conflate style and structure issues with consistency or citation problems. Isolating it as a standalone dimension improves both clarity and diagnostic value.

---

> ### Author Response · Authors · 2025-11-21
> **Response to Reviewer M3tr (2/5)**
>
> - **DeepScholar-Bench** focuses primarily on generating the *related work section* of academic papers, whereas our benchmark evaluates a model's ability to generate **long, open-ended research reports across diverse domains**.
>     - Their evaluation uses a single model (GPT-4.1 or GPT-4o) across all dimensions, with no human-verified rubrics. In contrast, we use a LLM-ensemble-as-a-judge with human-validated checklists.
>     - For Organization, DeepScholar-Bench conducts pairwise comparisons between generated and reference sections and reports a human–LLM agreement rate of 78%. Our checklist-based protocol achieves a higher agreement rate of 98.3%.
>     - For Coverage, their rubric is not verified, and their agreement rate is 72%, while ours is 100.0%.
>     - In addition, DeepScholar-Bench does not evaluate key aspects such as Analysis Depth, Factual & Logical Consistency, or Citation Traceability, all of which are central to our evaluation framework.
>
> We have included this expanded comparison and discussion in Appendix B of the revised manuscript.
>
> [2] https://arxiv.org/abs/2506.13784
>
> [3] https://arxiv.org/abs/2504.12516
>
> [4] https://arxiv.org/abs/2507.16280
>
> [5] https://arxiv.org/abs/2508.20033
>
> [6] https://arxiv.org/abs/2508.15804
>
> [7] https://arxiv.org/abs/2506.11763
>
> [8] https://arxiv.org/abs/2506.21506
>
> [9] https://arxiv.org/abs/2508.04183
>
> [10] https://arxiv.org/abs/2506.06287
>
> [11] https://arxiv.org/abs/2505.19253
>
>
>
> > W3: LLM-as-a-judge could be unreliable and expensive; more human agreement details needed
>
> Thank you for the thoughtful comment! Our evaluation design prioritizes both reliability and alignment with human judgment, guided by the following principles:
> - **Run-to-Run Stability**: Across 3 independent trials per dimension, system rankings remain unchanged. Raw score variances are also small (std 0.35-1.48).
> - **Strong alignment with human experts**: We selected an LLM ensemble (Gemini-2.5-Pro + GPT-5) based on empirical alignment:
>     - Presentation & Organization: Human study shows 98.3% agreement with either Gemini-2.5-pro or GPT-5
>     - Coverage & Comprehensiveness: Human 100.0% prefers Gemini or GPT-5 as judge; no instance uniquely favored Claude
>     - Analysis Depth: Human agreement with Gemini or GPT-5: 93.75%
>     - Factual & Logical Consistency and Citation Traceability:
>         - Additive (issue-based) scoring shows better human alignment and yields more stable ratings than direct scoring
>         - Factual & Logical Consistency: for identified issues, the human agreement rate is 82.0%
>         - Citation Traceability: for identified issues, the human agreement rate is 85.9%
>         - Our scoring penalizes every two issues, so final score alignment is even higher
>     - Citation Accuracy: Human agreement across flagged issues: 87.1%
> - **Judge Diversity**: We found relying on a single LLM to assign holistic scores (as used in [7]) for long-form open-ended reports introduces very high variance. As mentioned in our response to W2&Q4&W10, in our pilot study, if we use a single LLM to assign a score for analysis depth metric, the final performance can fluctuate by over 50 points across runs (e.g., 28.00 → 82.00). This motivated our use of:
>     - LLM-ensemble-as-a-judge to mitigate single model bias
>     - Dimension-specific protocols (e.g., checklist, additive, pairwise, rubric-tree) instead of directly assigning one single rating to enhance stability and interpretability
>
> We will expand these findings more explicitly in the revised paper.
>
> > W4: The paper defines six metrics but the main table averages only on four of them?
>
> Thank you for the thoughtful question! Our goal is not to reduce evaluation to a single aggregate score, but rather to provide a **multi-faceted assessment** that highlights the distinct strengths and weaknesses of each deep research system. As shown in Section 5, no system dominates across all metrics, collapsing all performance into a single rating risks obscuring important trade-offs.
> The average score was initially included for convenience. In the revised manuscript, we have removed the average score from the main table to avoid drawing undue attention.

---

> ### Author Response · Authors · 2025-11-21
> **Response to Reviewer M3tr (3/5)**
>
> > W5: How the 10 error patterns were selected? The risk of RL/finetuning leading to superficial gains on these patterns.
>
> Thank you for bringing this up! The 10 presentation-error patterns were derived through a rigorous LLM– and human-in-the-loop process focused on core qualities of well-written reports, such as clarity, structural coherence, correct citation usage, and consistent formatting, rather than superficial stylistic preferences. Our process was as follows:
> 1. A frontier model (GPT-5) was prompted to enumerate, identify and justify common presentation issues across generated reports from all 17 systems.
> 2. The resulting categories were clustered and consolidated; human experts were given access to report outputs, clustered categories, and then drafted the initial checklists.
> 3. Two independent human experts (with access to the same data) then independently added, removed, or modified checklist items, before discussing their rationale and iterating until consensus was reached.
> 4. The refined checklist was evaluated on a held-out subset of reports using GPT-5 and Gemini-2.5-Pro as judges to identify possible ambiguity or systematic misalignment.
> 5. Human experts then reviewed both the LLM-generated judgments and the raw reports. If systematic biases or failure cases were observed, the checklist was revised and steps 3–5 were repeated. This multi-round refinement was conducted three times to ensure robustness.
>
> Many items (e.g., matching in-text citations to references, enforcing a coherent document structure) require reasoning over the entire document rather than simple pattern memorization. Improving these behaviors through RL or finetuning typically requires stronger control over long-range coherence and formatting logic, not just superficial style adjustments.
> Because we do not collapse all metrics into a single score, performance on Presentation & Organization does not affect the evaluation of other dimensions. For future versions, we can apply the same process to update or diversify the checklists as deep research systems evolve.
>
> We will clarify this rationale and process more explicitly in the revised manuscript.
>
>
> > W6: Evaluating Factual & Logical Consistency at scale is a challenge for judge models
>
> Thank you for raising this point! To improve robustness and reduce variance in evaluation, we do not ask the LLM judges to directly assign consistency scores, which can vary significantly across runs. Instead, we find that models are more stable and reliable when tasked with identifying inconsistencies, rather than estimating an overall score. We therefore adopt an additive evaluation scheme: models identify factual or logical inconsistencies, and the final score is penalized by one point for every two issues detected.
> We also adopt additive scoring with an LLM-ensemble-as-a-judge (GPT-5 and Gemini-2.5-Pro), based on identified inconsistencies, as this approach yields stronger alignment with human judgments and more stable ratings compared to direct scoring. Across three independent runs, we observe that the ratings are highly stable, with small standard deviation (0.35).
>
> > W7: Analysis Depth: Looking at the prompt, it looks too heavy for the LMs to reason accurately over all the points.
>
> Thank you for pointing this out! Our goal is to obtain a **robust and reliable preference signal** over the analysis depth of generated reports. To make the evaluation task more feasible for the judge and improve evaluation stability, we decompose analysis depth into five interpretable sub-dimensions, assign scores for each, and aggregate them to compute a final score, then do pairwise comparison based on this calculated score, rather than asking the LLMs to directly choose a preferred report.
> We also employ LLM-ensemble-as-a-judge (GPT-5 and Gemini-2.5-Pro) and *mitigate positional and inductive bias* by switching the positions of the compared reports. Each comparison is evaluated four times (twice per judge, with positions swapped), and we report the averaged score.
> This setup results in substantially lower variance: across three independent runs, we observe **low score variance** (std 1.48). In contrast, if we ask an LLM to directly assign a single score or preference, the final scores can fluctuate by over 50 points across two independent runs, e.g., from 28.00 to 82.00, making the evaluation highly unstable and unreliable.

---

> ### Author Response · Authors · 2025-11-21
> **Response to Reviewer M3tr (4/5)**
>
> > W8.1: How claims are extracted from long reports and how accurate this extraction is. This step is crucial and should be evaluated itself given that the extraction could have been blocked by the websites.
>
> Thanks for raising this important point! We are happy to provide more details on the implementation.
> **Citation extraction**: For each report, we use Gemini-2.5-Pro to extract the (claim, URL) pairs. For stability, long reports are first split into smaller segments; extraction is performed on each segment, and the results are aggregated. The extraction step is repeated up to three iterations, allowing the model to refine and confirm completeness. This pipeline substantially improves reliability compared to prior work [7] (https://github.com/Ayanami0730/deep_research_bench/blob/main/utils/extract.py), which extracts all pairs using a single LLM call on the entire report.
> **URL content extraction**: As the reports to be evaluated are generated by agentic systems, none of the URLs contain pages that require a login or are behind a paywall. A key consideration is avoiding failures due to anti-bot detection. To this end, we have experimented with a few modern URL extraction tools:
> - Tavily (https://docs.tavily.com/documentation/api-reference/endpoint/extract)
> - Crawl4AI (https://github.com/unclecode/crawl4ai)
> - Gemini's URL context (https://ai.google.dev/gemini-api/docs/url-context)
> - OpenAI's web search (https://platform.openai.com/docs/guides/tools-web-search)
> Our pilot study shows that Gemini and OpenAI offer the highest reliability and cost efficiency, with zero anti-bot–related errors (e.g., HTTP 403) which we adopted for URL extraction.
>
>
> > W8.2: The rubric-tree framing of citation accuracy feels unnecessary for what is essentially a simple three-step pipeline
>
> Thank you for raising this important point! We agree that the **core workflow** can be summarized as a three-step pipeline: (1) identify each claim and its linked source, (2) check URL accessibility, and (3) assess whether the retrieved content supports the claim. However, a naive implementation of this simple pipeline becomes prohibitively slow and costly for papers containing dozens or hundreds of citations.
>
> Our rubric-tree framing is introduced primarily to improve efficiency, not to add conceptual complexity. It organizes two key design choices that significantly reduce the overhead:
>
> - Claim grouping: We group all claims that reference the same URL, requiring only a single web retrieval per group. For instance, if an average of four claims share a link across 100 citations, grouping reduces web requests by 4× (25 vs. 100), which translates to substantially fewer LLM calls and lower latency.
> - Relevance check: Before performing a full support assessment, we run a lightweight relevance check over the title or the first few hundred tokens of the webpage. If the link is clearly unrelated to the claim (e.g., a healthcare claim pointing to a technology-news article), we terminate the check early rather than processing the entire page. This avoids unnecessary extraction and long-context evaluation.
>
> > W9: Writing should be further polished. Some details are hard to comprehend and some details in the appendix appear to be made in a rush.
>
> We have revised and uploaded a new version of our manuscript based on your valuable feedback! The revised version includes the following updates and newly added sections:
>
> - Main paper: Expanded Sec. 4 DeepEval with more details on the motivation and design of evaluation protocol, human alignment studies. Refined Sec. 5 and removed the Avg column in Table 1 to avoid misunderstanding.
> - Appendix B: Expanded Related Works. Added a detailed comparison table across multiple dimensions, contrasting our benchmark with existing benchmarks and evaluation protocols.
> - Appendix C: Alignment of LLM Judges with Human Experts. Added a thorough description of our human-alignment study, including procedures, metrics, and results for each evaluation criterion.
> - Appendix D: DEERFLOW+. Introduced DEERFLOW+, an enhanced variant of the original DEERFLOW framework, addressing frequent failure modes, adding inline-citation support, and robust long-context management.
> - Appendix E: Evaluating Citation Accuracy with a Rubric Tree. Improved the description and visualization of the rubric-tree framework, with additional details on optimization strategies and citation-error analysis.
> - Appendix G: Error Pattern Examples. Expanded this section with new representative failure cases, such as embedded citations breaking text flow and hallucinated references.
> - Appendix H: Annotation Team. Added further details on annotator backgrounds.
> - Appendix J: Human Verification Guidelines. Added comprehensive guidelines describing our human-verification protocol and step-by-step procedures.
> - Appendix K: Results Breakdown. Added a per-criterion breakdown of agentic-system performance across all ten presentation checklist criteria.

---

> ### Author Response · Authors · 2025-11-21
> **Response to Reviewer M3tr (5/5)**
>
> > Q1: What is included in the labor investment?
>
> The reported time investment refers to human labor spent on curating the benchmark and designing the evaluation protocols. This includes:
> - The full 11-step benchmark construction and verification pipeline described in Figs.3&4 (e.g., crowd study, task construction, expert validation, quality control)
> - Additional effort for evaluation protocol design, including rubric and checklist creation, multi-round refinement (e.g., human labor spent to construct checklists for presentation in our response to W5), quality control, and human–LLM alignment studies
>
> This estimate reflects only the time spent on human dataset and evaluation protocol curation, annotation, and validation. It does not include time spent on code implementation or infrastructure development.
> We will include a more detailed breakdown of the time allocation in the revised manuscript.
>
> > Q2: DeepEval appears to involve many LLM calls. What is the end-to-end cost and latency to evaluate a single system on the full benchmark?
>
> Thank you for raising this point! Evaluating long-form, open-ended reports reliably is inherently challenging, and we have designed our rubric-tree framework specifically to improve efficiency on citation evaluation. Our implementation also leverages multiprocessing and asynchronous I/O to maximize concurrency and reduce overhead.
>
> With our implementation, evaluating 100 reports takes less than 20 minutes for each of the five metrics. Citation accuracy takes longer as its runtime scales with the number of citations and the grouping level (the number of citations aggregated together).
>
> While the evaluation pipeline itself is optimized and runs efficiently, the end-to-end latency experienced by a user depends on several external factors beyond our control, including the choice of third-party LLMs (e.g., Gemini-2.5-Pro, GPT-5), network conditions, and provider-specific rate limits. Similarly, the monetary cost varies with the user’s organizational tier, negotiated rates, and available discounts from the LLM service providers. There are multiple ways to further reduce the cost if necessary (e.g., switch models, adjust max retrial, or use Batch API). We will include a brief discussion of these practical considerations in the revised version to clarify expected variability.
>
> >Q3: Will the benchmark be fully open sourced? (including the tasks, task-specific checklists, code, etc.)
>
> Yes, we have received the institutional approval and the whole benchmark (including tasks, checklists, and code) will be fully open-sourced and maintained to support future research on deep research agents and the evaluation of long-form, open-ended outputs.
>
> > Q5: OpenAI Deep Research seems more like a multi-agent system, given the information revealed?
>
> Based on the documentation available to us [12][13], we do not currently see any evidence suggesting that OpenAI Deep Research is a multi-agent system. If OpenAI later provides information indicating that it uses a multi-agent architecture, we would be happy to update the categorization.
>
> [12] https://cookbook.openai.com/examples/deep_research_api/introduction_to_deep_research_api
>
> [13] https://platform.openai.com/docs/guides/deep-research

---

> ### Comment · Reviewer_M3tr · 2025-11-21
>
> Good rebuttal! thank you for the hard work.
>
> While I may not agree with some of the details and still have concerns about the benchmark design and experiments, especially when contextualized with the surge of recent deep research benchmarks, I will raise the score to 6 since the response clearly reflects substantial effort of both the rebuttal and the project.
>
> I would also suggest always preparing a more polished and **detailed** initial manuscript in the future to help reduce the number of clarification questions that reviewers have to ask, especially given the complexity of this paper (benchmark design and experiments). This is a serious problem for an initial submission.

---

> > ### Author Response · Authors · 2025-12-03
> >
> > Dear Reviewer `M3tr`,
> >
> > Thank you for your thoughtful feedback and support of our work! We greatly appreciate your constructive insights, particularly regarding evaluation design and overall organization. We have incorporated these improvements into our revised manuscript. Your suggestions have meaningfully strengthened the clarity and quality of our manucript.
> >
> > Best regards,
> >
> > Authors

---

### Official Review · Reviewer_m18a · 2025-11-04

**Soundness:** 3
**Presentation:** 3
**Contribution:** 3
**Rating:** 8
**Confidence:** 3

**Summary:**

This paper addresses LiveResearchBench, a benchmark of 100 expert-curated, time-varying tasks designed around four principles: user-centricity, unambiguous scope, temporal dynamism, and multi-faceted reasoning. The authors further propose DeepEval, an evaluation suite measuring presentation, factual consistency, coverage, analytical depth, and citation reliability through rubric-tree scoring and ensemble LLM-as-a-Judge evaluation. Evaluations on 18 single- and multi-agent systems show existing models excel at information collection but fall short in analytical depth and citation faithfulness.

**Strengths:**

(1) The paper is well-motivated by the lack of realistic, dynamic benchmarks for deep research. LiveResearchBench is constructed with clear design principles—user-centric, unambiguous, time-varying, and multi-faceted—and backed by a solid six-stage curation and five-stage verification process. The use of clarification questions, checklist-based unit tests, and expert validation ensures task clarity and evaluation consistency.

(2) DeepEval provides a six-dimensional framework that goes beyond accuracy to assess coverage, analytical depth, and citation reliability. The rubric-tree and checklist design make LLM-as-a-Judge both scalable and interpretable, achieving strong alignment with human evaluation.

(3) The evaluation and analysis are comprehensive, offering insights into the current bottlenecks and future directions for deep research agents.

**Weaknesses:**

One of the benchmark’s core strengths—its dynamic, time-varying nature—also poses practical challenges. Time-sensitive webpages and evolving content make long-term maintenance difficult. It would be helpful if the authors could provide evidence or analysis showing the benchmark’s stability and validity over time.

**Questions:**

(1) How stable are the system rankings across repeated runs or different judge ensembles?
(2) Could you provide inter-judge agreement or human–LLM agreement for each evaluation dimension separately?

---

> ### Author Response · Authors · 2025-11-21
>
> Thank you for your thoughtful and constructive feedback, and for recognizing the motivation, design rigor, and evaluation depth of our benchmark and evaluation framework! We address your comments point by point below:
>
> > W: Time-varying nature, benchmark stability and validity.
>
> Thank you for your insightful feedback! While LiveResearchBench is designed to reflect real-world, time-varying research scenarios, we carefully mitigate concerns about long-term validity through both query design and rigorous validation.
> Our task design emphasizes **semantic stability**: although surface-level search results may evolve, the underlying reasoning challenges remain well-posed and answerable regardless of time. For example, as shown in Figure 1, a task may reference trends "up to {{current date}}", but its core intent, e.g., analyzing cross-cultural developments in art, is temporally resilient.
>
> Moreover, our five-stage verification pipeline ensures robustness: each task and checklist undergoes ambiguity resolution, expert review, rubric testing, and quality control. We also periodically re-verify the benchmark. To date, no task has become obsolete or unanswerable. We have now included this discussion in the revised manuscript.
>
> > Q1&Q2: Stability of system rankings and inter-judge/human–LLM agreement across evaluation dimensions
>
> Thank you for raising this important point! The selection of the LLM-ensemble-as-a-judge (Gemini-2.5-Pro + GPT-5) and the design of each dimension-specific evaluation protocol are guided by both **run-to-run stability** and **high alignment with human experts**:
> - Ranking Stability: For each model and each evaluation dimension, we conducted 3 independent runs using our final LLM ensemble (Gemini-2.5-Pro + GPT-5). Rankings remained unchanged across all runs.
> - Judge Ensemble Selection: We evaluate with Gemini-2.5-Pro, GPT-5, and Claude-4-Sonnet. The Gemini-2.5-Pro + GPT-5 ensemble consistently achieves the highest human agreement and mitigates single-model bias, so we adopt it as the default.
>
> We have included the detailed statistics and alignment studies in the revised manuscript and provide an overview below:
> - Presentation & Organization:
>     - Agreement across all 3 LLMs (Gemini-2.5-Pro, GPT-5, and Claude-4-Sonnet): 67.8%
>     - Agreement across Gemini and GPT-5: 83.6%
>     - Human study shows 98.3% agreement with either Gemini or GPT-5
> - Coverage & Comprehensiveness:
>     - Human 100.0% prefers Gemini or GPT-5 as judge; no instance uniquely favored Claude
> - Analysis Depth: Human agreement with Gemini or GPT-5: 93.75%
> - Consistency & Citation Traceability:
>     - Pointwise additive (issue-based) scoring shows better human alignment and yields more stable ratings than direct scoring.
>         - Consistency: for identified issues, the human agreement rate is 82.0%
>         - Citation Traceability: for identified issues, the human agreement rate is 85.9%
>     - Note: our default weighting scheme decreases the score for every two issues detected. Therefore, the actual alignment of final scores with human judgment would be even higher.
> - Citation Accuracy: Human evaluators agree with either Gemini or GPT-5 on 87.1% of (claim, URL) support judgments.

---

### Author Response · Authors · 2025-12-04
**General Response and Summary of Discussions**

Dear Program Chairs, Senior Area Chairs, Area Chairs, and Reviewers,

We would like to express our sincere gratitude for your thorough and insightful review of our submission. We deeply appreciate your constructive feedback and your recognition of the significance of both our benchmark and our evaluation methodology. Reviewers highlighted several core strengths of our work: the realistic and dynamic benchmark design and its rigorous multi-stage curation and verification pipeline (m18a, Mkd1, M3tr), which they noted as a convincing improvement over prior work (M3tr, Mkd1); the breadth, interpretability, and reliability of the DeepEval suite, including its six-dimensional metrics and the associated checklist-, additive-, pairwise-, and rubric-tree–based evaluation protocols with strong human alignment (m18a, M3tr, Mkd1); and the comprehensive evaluation across frontier deep-research agents, which surfaced meaningful insights into current system limitations (m18a, M3tr, wzms, Mkd1).

During the rebuttal period, we provided detailed responses to reviewers’ questions and incorporated meaningful improvements inspired by their suggestions. **We also appreciate Reviewer M3tr’s positive acknowledgment of our rebuttal and the substantial effort invested, and are encouraged by the score increase to 6.** A summary of the discussion is provided below:

### 1. **Benchmark Stability and Time-Varying Validity**

Responding to Reviewer m18a, we expanded our discussion of how LiveResearchBench handles temporal dynamism by emphasizing the *semantic stability* of tasks and checklist design, and by clarifying why tasks remain well-posed even as surface-level web content evolves.

### 2. **LLM-Ensemble-as-a-Judge, Evaluation Protocols, and Human–LLM Agreement**

Following Reviewers m18a, M3tr, and Mkd1, we added full details on ranking stability across repeated runs and reported *per-dimension* human–LLM agreement statistics for all six evaluation metrics in Appendix C. We also detailed the rationale behind the design of our four evaluation protocols and the selection of our LLM-ensemble-as-a-judge (Gemini-2.5-Pro + GPT-5), including why ensemble judging is preferred and why Claude-4-Sonnet was not included in the final ensemble.

We additionally elaborated on the construction of checklists and rubric-tree evaluators (Reviewer M3tr), explaining the multi-round human-in-the-loop refinement process and the reasons for using checklists, additive, issue-based scoring instead of single holistic scores. These clarifications illustrate the robustness of our evaluation pipeline and the strong human alignment achieved across dimensions, ensuring reliability for assessing long-form open-ended research reports.

### 3. **Expanded Comparisons to Prior Benchmarks and Evaluation Methods**

Addressing Reviewers M3tr and wzms, and building on Reviewer Mkd1’s comments, we added a detailed comparison to prior benchmarks in Appendix B. This includes a summary table contrasting LiveResearchBench with representative deep-research and search benchmarks (e.g., DeepResearch Bench, ResearcherBench, DeepScholar-Bench, Mind2Web2) across key dimensions such as human-verified rubrics, expert-curated tasks, open-ended tasks, multi-domain coverage, dynamic queries, and use of LLM ensembles.

We also elaborated on how previous benchmarks evaluate long-form outputs and how our six-dimensional DeepEval framework addresses their limitations through broader metric coverage, human-verified rubrics, and higher human-alignment rates.

### 4. **Clarified Task Design, Scope, and Practicality**

In response to Reviewers M3tr and wzms, we clarified that our benchmark uses *final clarified research prompts*, reflecting realistic usage of deep-research systems after their clarification turns, ensuring consistent interpretation both within and across agentic systems. Following Reviewer Mkd1, we also refined the explanation of “wide information search”, highlighting tasks that require cross-source retrieval, multi-constraint filtering, and synthesis.

We have addressed all concerns raised by the reviewers and updated the manuscript accordingly. **LiveResearchBench and DeepEval together offer a realistic, rigorously validated, reliable, and multi-faceted benchmark and evaluation framework for evaluating deep research agents.**

We sincerely thank the Program Chairs, Senior Area Chairs, Area Chairs, and Reviewers for your time and dedication in reviewing our submission.

Best,

Authors of Submission 10102

---

### Meta-Review · Area_Chair_mDQ6 · 2026-01-05

**Summary:**

The paper introduces LiveResearchBench, a benchmark of 100 expert-curated, dynamic, user-centric tasks requiring real-time web search and synthesis into citation-grounded long-form reports. It also proposes DeepEval, a six-dimensional evaluation suite using checklists, rubric-trees, additive scoring, and LLM-ensemble judging to assess presentation, coverage, consistency, analysis depth, and citation quality. Extensive evaluation of 18 frontier single- and multi-agent systems reveals strengths in information gathering but persistent weaknesses in analytical depth and citation reliability.

Reviewers praised the realistic task design emphasizing unambiguity and temporal dynamism, the rigorous multi-stage curation and verification pipeline, the breadth and interpretability of DeepEval (with strong human-LLM alignment), and the comprehensive analysis yielding actionable insights. Initial concerns included benchmark maintenance over time, reliance on LLM judges, clarity of metric design and comparisons to prior work, potential over-prescription of tasks, and evaluation cost/latency.

The authors' detailed rebuttal and manuscript revisions effectively addressed these issues: expanded discussions on temporal stability, added per-dimension human-LLM agreement statistics and ranking stability evidence, provided detailed benchmark comparisons in tables, clarified task scope reflecting real-world clarified queries, refined metric motivations, and committed to full open-sourcing. One reviewer explicitly raised their score in recognition of the thorough response.

**Reviewer Concerns:**

Nearly all concerns were resolved through clarifications, new appendices (e.g., comparison tables, alignment studies, refined protocols), and manuscript updates. Minor remaining points on long-term maintenance and evaluation cost are acknowledged as practical considerations rather than flaws.

**Reviewer Scores:**

Initial scores ranged from marginally below (4) to accept (8), with positive recognition of rigor and impact. Post-rebuttal engagement led to at least one upward adjustment (to 6).

---

### Decision · Program_Chairs · 2026-01-26

Accept (Poster)